# SIFT-MS optimization for atmospheric trace gas measurements at varying humidity

Ann-Sophie Lehnert[1,2], Thomas Behrendt[1], Alexander Ruecker[1], Georg Pohnert[2], Susan E. Trumbore[1]

[1]Department of Biogeochemical Processes, Max Planck Institute for Biogeochemistry, 07745 Jena, Germany
[2]Institute for Inorganic and Analytical Chemistry, Friedrich Schiller University, 07743 Jena, Germany

*Correspondence to*:

Ann-Sophie Lehnert (alehnert@bgc-jena.mpg.de)

Department of Biogeochemical Processes

Max Planck Institute for Biogeochemistry

Hans-Knoell-Straße 10, 07743 Jena, Germany

For submission to: Atmospheric Measurement Techniques

25

**Abstract.** As direct real-time analysis techniques, Selective Ion Flow Tube Mass Spectrometry (SIFT-MS) and Proton-Transfer Reaction Mass Spectrometry (PTR-MS) provide on-line measurement of volatile organic compounds (VOCs). Both techniques are widely-used across several disciplines, *e.g.* atmospheric chemistry, food science and medicine. However, the humidity of the sampled air greatly influences the quantified mixing ratio, and must be accounted for. Here we present several improvements to a Voice 200 ultra SIFT-MS instrument to reduce background levels and enhance sensitivity. Increasing the sample gas flow to 125 sccm enables LODs at sub-ppb level, and the resulting humidity-dependence is overcome by calibrating for humidity as well. A comparison with a PTR-QMS 500 showed detection limits of the PTR-MS still being an order of magnitude lower, whereas sensitivity was higher for SIFT-MS and its calibration was still more robust against humidity. Thus, SIFT-MS is a suitable, lower-cost and easy-to-use alternative for atmospheric trace gas measurements of more complex mixtures, even with isomers, at a varying humidity range.

## Introduction

Volatile organic compounds (VOCs) shape the medium we are living in: the air. As odours, pheromones, reductants, greenhouse gases and precursors for aerosols, they regulate key processes in the environment. Due to their reactivity, their atmospheric lifetimes are usually limited, and their mixing ratios are rather low and span several orders of magnitude, typically tens of parts per trillion (ppt) – low parts per million (ppm). Despite great improvements during the past years, methods of measuring VOC that rely on concentrating samples using adsorption tubes, or trapping air in storage containers often have artefacts due to dissipation of the analytes to, or reactions with, the walls, sorptive materials or tubing used in experimental setups (Herrington, 2015, Piennar et al., 2015, Deming et al., 2019).

Thus, an easy, fast, and direct analysis method is desirable. Proton-Transfer Reaction Mass Spectrometry (PTR-MS) and Selective Ion Flow Tube Mass Spectrometry (SIFT-MS) both provide these characteristics as they do not rely on time-consuming sample separation like Gas Chromatography/Mass Spectrometry. Both are used in a wide variety of fields comprising both natural and anthropogenic atmospheric chemistry (Milligan *et al.*, 2002; Yuan *et al.*, 2017), plant studies (Amelynck *et al.*, 2013), food science (Davis *et al.*, 2005), and medical applications like breath-analysis (Schwarz *et al.*, 2009; Shende *et al.*, 2017; Smith *et al.*, 2014).

The two techniques have been compared in various reviews, *e.g.* Bylinski, *et al.* (2017), Casas-Ferreira *et al.* (2019), Smith and Spanel (2011), and therefore, SIFT-MS and its main differences to PTR-MS are only discussed briefly here. The principle behind both instruments is the chemical ionization of the analyte during a defined reaction time. Thus, the amount of compound can be calculated from the number of detected product ions using the kinetic rate constants $k$ of the ionization reaction of the analyte A with the reagent ion $R^+$:

$$A + R^+ \rightarrow A^+ + R \tag{R1}$$

$$\frac{d[R]}{dt} = k \cdot [A] \cdot [R^+] \tag{1}$$

Assuming a pseudo-first order reaction with $[R^+] \ll [A]$, the differential equation can be solved by an exponential decay function (McEwan, 2015), and using theoretical knowledge of diffusion behaviour and gas and ion velocities in an electric field as well as experimental factors correcting for mass discrimination, one can estimate the analyte concentrations (Smith and Spanel, 2005). Since SIFT-MS uses a flow tube that transports the ions through the gas flow, and only uses a small voltage to minimize diffusion to the walls, near-thermal conditions apply unlike in PTR-MS and mixing ratios can be determined with an accuracy of ±35 % (Langford *et al.*, 2014).

Both instruments are comprised of the same three components: an ion generation zone, a reaction zone and a detection zone, cf. Fig. 1 for a scheme of SIFT-MS. Reagent ions $H_3O^+$ (both instruments), $NO^+$ and $O_2^+$ (SIFT-MS only, with positive ion source) are generated and injected into the reaction zone, where they chemically ionize the analytes to form product ions, *e.g.* Eq. (R2) for methanol:

$$H_3O^+ + CH_3OH \rightarrow H_2O + CH_3OH_2^+ \qquad\qquad (R2)$$

All ions are then analysed by a mass spectrometer (MS), usually a quadrupole-MS for SIFT-MS and a time of flight-MS for PTR-MS, separating the ions by their *m/z* ratio and then counting the number of ions hitting the multiplier.

There are two main differences between the two instruments: First, they differ in the way the reagent ions are generated and second, whether the ions are reacting with the analyte in a drift-tube vs. in a flow-tube. Whereas PTR-MS uses hollow-cathode discharges to ionize water vapour generating $H_3O^+$ (Romano *et al.*, 2015), SIFT-MS generates a wet air plasma *via* microwave discharge and then selects the reagent ions $H_3O^+$, $NO^+$, and $O_2^+$ with a quadrupole (Smith and Spanel, 2005). Since the three reagent ions react differently with the analyte and may form different association and fragmentation products, more structural information can be obtained. However, the efficiency of creating the reagent ions is lower than for PTR-MS, leading generally to higher limits of detection (LOD) for SIFT-MS.

The SIFT-MS uses a flow tube with an inert carrier gas (He or $N_2$) that is mixed with the sample gas containing the analyte and a low voltage to focus the ions, whereas PTR-MS uses a drift tube through which the ions are guided and accelerated by a much higher electric field. Due to collisions with the carrier gas, in SIFT-MS the analytes and reagent ions are approximately in thermal equilibrium. Because of their acceleration, the effective temperatures of the ions in the tube are much higher for PTR-MS than for SIFT-MS, and these differences in energy lead to different fragmentation patters for the two methods (Biasioli *et al.*, 2011). The carrier gas needed in SIFT-MS serves the additional role of reducing the amount of ion clustering, like water clustering (*e.g.* $H_3O^+ \cdot H_2O$ or $CH_3OH_2^+ \cdot H_2O$) that can occur at high humidity.

As mentioned above, the high limits of detection (LODs) for SIFT-MS can be an issue when measuring atmospheric trace gases, so we optimized the Voice 200 ultra SIFT-MS (Syft Technologies, New Zealand) to reach sub-ppb LODs and systematically characterized the performance of the SIFT-MS under different humidity-conditions. Lastly, the instrument's performance was compared to the performance of a PTR-QMS 500 (Ionicon, Austria).

**Experimental section**

**Materials**

VOC-free air was generated by a Pure Air Generator (PAG 003, Ecophysics, Dürnten, Switzerland), and was further purified by a scrubber built into a Gas Calibration Unit (GCU, Ionicon, Austria). Gas mixtures of known VOC mixing ratios were produced by diluting a VOC standard gas mixture (Ionicon, Austria) (1 ppm each of 2-butanone, acetaldehyde, acetonitrile, acrolein, benzene, chlorobenzene, crotonaldehyde, dichlorobenzene, ethanol, isoprene, methanol, α-pinene, toluene, *o*-xylene in nitrogen). The GCU was used to dilute the standard to the mixing ratios used in the calibration. To minimize background, the tubing used was 1/8" black PFA tubing with 1/8" Swagelok stainless steel connectors. Sample gas fluxes through the multiport-inlet system were measured *via* a Sensidyne Gilian Gilibrator-2 NIOSH Primary Standard Air Flow Calibrator (Sensidyne, FL, USA).

**SIFT-MS optimization**

Hardware and parameters were changed to optimize the Syft Voice 200 ultra with a positive ion source and a multi-port inlet. The shut-down valve in the carrier gas line was removed upon the advice of Marvin Shaw (University of York, GB). We also removed the vent valve for the backing pumps and just vent the system through the flow tube with purified air. All

the Viton/FKM and nitrile o-rings delivered with the instrument were replaced by Hennlich FEP-coated FKM o-rings. Further, the VICI-valve that was delivered with the multi-port inlet system was switched to a flow-through VICI valve (EUT-6CSF16MWE).

VICI silica-coated stainless steel capillaries with capillary sizes of 0.007", 0.010", and 0.015" inner diameter (ID) as well as PEEK capillaries (BOLA S1817-08, 0.25 mm ID, Bohlender, Germany, ChromaTec, 0.3 mm ID, Labomatic Instruments AG, Germany, PEEK Capillary Tubing 37010-20, 0.010" ID, Thermo Scientific, USA, and Latek Blue PEEK capillaries 8560 – 6009, 0.25 mm ID, Latek, Germany) as well as a Swagelok SS-SS2 needle valve were tested as inlet capillaries. For both dry and humid VOC-free air (90 % humidity at 25°C), a background was measured between $m/z$ = 15 and 250 u (100 ms count time per ion, 10 scans). The background was normalized to both $10^6$ counts of the respective reagent ion and the flowrate through the inlet capillary, see Fig. S1.

Microwave cavity and power, the upstream and downstream lenses, the source pressure and the air stream into the source were tuned before the measurements. Flow tube voltage and temperature, and carrier gas flow were optimized for VOCs with mixing ratios lower than 10 ppb. These experiments were performed using both helium and nitrogen as carrier gases, see Fig. S3-S14. Each time, 5 ppb of the VOC standard was mixed into dry and humid (90 % relative humidity at 25°C) VOC-free air. The flow tube voltage was scanned in 5 V steps between 0 and 65 V, the flow tube temperature was stepwise increased in 5°C intervals from 100 to 160°C, and the carrier gas flow was scanned at 0, 7.89, 15.79, 31.57, 47.36, 63.14, 78.93, 118.39, 157.85, 236.78, 315.71, 394.63, and 473.56 ccm (0-6 TorrL s$^{-1}$). For the scan, 15 scans were conducted with 500 ms dwell time – the time the detector integrates the signal – per ion after 20 s settle time.

To select for nitrogen versus helium as a carrier gas, calibrations were done in the range from 0.1 and 10 ppb for the VOC standard in dry air as well as at 30 %, 60 %, and 90 % relative humidity (25°C). For the measurement, after 20 s settle time, 15 scans were conducted with 500 ms dwell time per ion, except for α-pinene masses m/z = 81 and 137 u (H$_3$O$^+$ reagent ion), which were measured for 1 s, to account for its low mixing ratio due to its semivolatility in our soil samples.

**Evaluation of different calibration procedures**

The instrument calibration done with helium carrier gas (see Sect. 2.2) was used for evaluating different calibration procedures. Different regression equations and calibration procedures were tested: In the following equations, $I_P$ is the product ion intensity, $I_R$ is the reagent ion intensity, $\chi$ is the mixing ratio of the analyte, $\phi$ is the relative humidity, $ICF$ is the experimentally determined instrument calibration factor the SIFT-MS provides for correcting discrimination effects in flow tube and downstream quadrupole, $k$ is the kinetic rate constant, $I_{H_3O^+}$ is the intensity of the H$_3$O$^+$ ion, $I_{H_3O^+ \cdot H_2O}$ the intensity of the H$_3$O$^+ \cdot$H$_2$O ion, and $m, a, b, c,$ and $d$ are regression parameters that are fitted. In the equations where more than one reagent and product ion was included (*e.g.* water clusters of product ions), the different ions were indexed by $i$ and $j$.

1. Calibration for each humidity

    a. absolute product ion intensities: $I_P = m \cdot \chi_P + c$     (2)

    b. relative product ion intensities: $\frac{I_P}{I_R} = m \cdot \chi_P + c$     (3)

2. Calibration with linear humidity-dependence:

    a. Absolute product ion intensities: $\chi = m_1 \cdot I_P + m_2 \cdot \phi + b$     (4)

    b. Relative product ion intensities: $\chi = m_1 \cdot \frac{I_P}{I_R} + m_2 \cdot \frac{I_{H_3O^+}}{I_{H_3O^+ \cdot H_2O}} + b$     (5)

3. Based on the instrument's concentration result: $\chi_{substance} = \chi_{measured} \cdot \frac{k_1 \cdot I_{H_3O^+} + k_2 \cdot I_{H_3O^+ \cdot H_2O}}{c_1 \cdot k_1 \cdot I_{H_3O^+} + c_2 \cdot k_2 \cdot I_{H_3O^+ \cdot H_2O}}$     (6)

4. Calibration derived from physical parameters:

    a. Completely de novo: $\chi_{substance}(ppbv) = a \cdot \frac{I_{P_1^+} + \Sigma_{i=2}^{N}\left(b_i \cdot I_{P_i^+}\right)}{I_{R_1^+} + \Sigma_{j=2}^{M}\left(c_j \cdot I_{R_j^+}\right)} + d$     (7)

    b. Using the instrument calibration function: $\chi_{substance} = a \cdot \frac{\Sigma_i\left(I_{P_i^+} \cdot ICF_{P_i}\right)}{I_{R_1^+} \cdot ICF_{R_1} + \Sigma_j\left(b_j \cdot I_{R_j^+} \cdot ICF_{R_j}\right)} + c$     (8)

    c. De novo with relative values derived from Eq. (7): $\chi_{substance}(ppbv) = a \cdot \frac{\frac{I_{P_1^+}}{I_{R_1^+}} + \Sigma_{i=2}^{N}\left(b_i \cdot \frac{I_{P_i^+}}{I_{R_1^+}}\right)}{1 + \Sigma_{j=2}^{M}\left(c_j \cdot \frac{I_{R_j^+}}{I_{R_1^+}}\right)} + d$     (9)

From the raw data taken at each calibration point, the five datapoints before the last datapoint were used for the regression to minimize the effect of instable flows. Based on the blank measurement, the critical intensity was calculated by Eq. (10).

$$I_{crit} = \bar{I}_{Blank} + 3 \cdot sd(I_{Blank}) \tag{10}$$

Only calibration points with means above the critical value were included in the regression. The evaluated ions are shown in Table S4. For the sake of simplicity, we will refer to the individual ions by $m/z$(reagent ion) / $m/z$(product ion) / analyte, *e.g.* 19 u / 33 u / methanol throughout the paper.

To assess the quality of the regression models, the Bayesian Information Criterion was calculated for each regression of the different compounds, see Eq. (11). Based on the variance of the residuals, it gives a measure of how well the model fits – a lower value shows a better fit of the model. In comparison to Akaike's Information Criterion, it more strongly punishes a higher number of parameters (Veres, 1990).

$$BIC = n \log\left(\widehat{\sigma_R}^2\right) + k \log(n) \tag{11}$$

$n$… number of samples, $\widehat{\sigma_R}^2$… variance of the residuals, $k$ … number of model parameters.

The BICs were calculated individually for each compound, but to get an overall idea on how the regression functions perform, mean, median, maximum and minimum of the BIC values of the compound obtained for each method were compared, see Table S5.

For the comparison of the SIFT-MS with the PTR-MS, each humidity was compared separately from the others following a basic calibration function, see Eq. (12).

$$\frac{I_{product\ ion}}{I_{reagent\ ion}} \cdot 10^6 = m \cdot [analyte] + c \tag{12}$$

The limit of detection (LOD) was estimated to be three times the standard deviation of the blank. The sensitivity was defined as the change in signal response by mixing ratio change, i.e. the slope of the respective calibration function. The confidence interval (CI) of the sensitivity was calculated as Eq. (13).

$$CI_{m,95\%} = t_{(p=95\%,df=26)} \cdot \frac{s_{y,x}}{\sqrt{SS_{xx}}} \tag{13}$$

$t_{(p=95\%, df=26)}$ is the 95% value of Student's t-distribution for 26 degrees of freedom, $s_{y,x}$ the residual standard deviation, and $SS_{xx} = \sum_i (x_i - \bar{x})^2$ the sum of squares of the mixing ratios.

The signal to noise ratio was calculated by dividing the normalized product ion intensity at 1 ppb standard gas by the normalized product ion intensity of the blank (no VOC standard), Eq. (14).

$$SNR = \frac{I_{1ppb}}{I_{Blank}} \tag{14}$$

For the signal to noise ratio, upper and lower CIs were calculated separately, the upper CI by Eq. (15), the lower CI as Eq. (16), where $sd()$ is the standard deviation of the respective intensity.

$$CI^u_{SNR,95\%} = t_{p=95\%, df=7} \cdot \left( \frac{I_{1ppb} + sd(I_{1ppb})}{I_{Blank} - sd(I_{Blank})} - SNR \right) \tag{15}$$

$$CI^l_{SNR,95\%} = t_{p=95\%, df=7} \cdot \left( SNR - \frac{I_{1ppb} - sd(I_{1ppb})}{I_{Blank} + sd(I_{Blank})} \right) \tag{16}$$

**Comparison of SIFT-MS and PTR-MS**

The SIFT-MS was compared to a PTR-QMS 500 (Ionicon, Austria) by calibrating both instruments in the same manner as Sect. 2.3. For the calibrations, 10 measurements were performed at each mixing ratio for each level of humidity. For both instruments, the ion dwell time was set to 500 ms to ensure comparability. The α-pinene masses $m/z = 81$ and 137 u ($H_3O^+$ reagent ion) were measured for 1 s. The masses measured for the different compounds can be found in Table 1. The counts were normalized to $10^6$ counts of the reagent ion. The PTR-MS was operated at E/N = 136 Td (inlet temperature 85°C, drift tube temperature 60°C, drift tube voltage 600 V, drift tube pressure 2.25 mbar), and the counts of $m/z = 19$ u were inferred from its isotopic peak, $m/z = 21$ u.

**SIFT-MS robustness over time**

To test the SIFT-MS robustness over time, we did three calibrations as described in Sect. 2.3 for 60% humidity on one day (day 1) and repeated this one week later (day 8). All calibration curves were fitted with a linear regression. The significant difference of the slopes and intercepts of the two days was tested using an F-test (p = 95%, Bonferroni-corrected to 99.86%, n = 37) and depending on the result of the F-test, the homogeneous-variance or heterogeneous variance t-test (p = 95%, Bonferroni-corrected to 99.86%, for the correction n = 37) was applied. In addition to that, the 2 ppb calibration points from day 1 and day 8 were compared using a Bartlett test and an ANOVA (p = 95%, Bonferroni corrected to 99.935%, for the correction n = 77). Their relative standard deviation was calculated.

To evaluate a longer time scale, the measurement of a standard gas mixture of benzene, o-xylene, octafluorotoluene, hexafluorobenzene, ethylene, isobutane, tetrafluorobenzene, and toluene (2ppm each in nitrogen, Syft, New Zealand) on each working day was evaluated. A Neumann trend test was used to test for trends (p = 95%, n = 10).

To see the effect of venting the instrument on the calibrations, e.g. for maintenance or reparations, calibrations done in May 2018, December 2018, and January 2019, before and after the o-ring change and a detector shutdown, were conducted and compared as described above.

**Results and discussion**

Complete results of the different combinations of humidity conditions, carrier gas, flow tube temperature and voltage are given in the supplemental material; here we show selected comparisons under a subset of experiments and conditions that best illustrate the performance of the SIFT-MS and how it compares to the PTR-MS.

## SIFT-MS-optimization to improve sensitivity

Several changes were applied to the SIFT-MS to improve its limit of detection, inspired by the parameter optimizations done by Marvin Shaw (University of York, unpublished results), but considering different sample humidities: Amongst others, the inlet capillary was replaced by a needle valve (see Fig. S1) and all the o-rings in the instrument were replaced with FEP-coated FKM o-rings (see Fig. S2). All measures led to a significant reduction of the instrument background, by up to factor 5 for some masses (see Fig. S2).

Besides the hardware changes, we also optimized a number of running parameters, including the flow tube voltage, flow tube temperature, carrier gas flow, and sample gas flow. The observed effects of water clustering, adduct formation, fragmentation and humidity-sensitivity match the theoretical considerations of Smith and Spanel, 2005:

As was expected, the product ions increase with increasing sample gas flow in most cases (see Fig. 2 or Fig. S3 and S4 for complete results), and also water clustering increases. However, we were surprised to see that the effect of water clustering was not critical for the chosen settings: the amount of unreactive $H_3O^+ \cdot 2H_2O$ ($m/z = 55$ u) was negligible, and no $H_3O^+$ signals were visible in the other two reagent ion channels. For methanol, one can already observe the effect of an increased amount of $H_3O^+ \cdot 2 H_2O$ with increasing sample gas flow, leading to less background on $m/z = 33 u$ as also shown by de Guow and Warneke, 2007. However, this experiment was performed at a medium humidity with a sample gas flow of 125 sccm. To further decrease the amount of $H_3O^+ \cdot H_2O$ formed, flow tube voltage and temperature as well as the carrier gas flow were also optimized.

With higher flow tube voltage, *i.e.* a higher kinetic energy of the ions, we expected (i) a higher reaction efficiency in general, leading to more ions, (ii) more secondary reagent ions, e.g. more $H_3O^+$ when $O_2^+$ was the reagent ion, (iii) less water clustering, (iv) less adduct formation and (v) more fragmentation. In Fig. 2 (b) (Fig. S6 for all ions), one can see that (iii) and (iv) are definitely true, (v) does occur a bit, but hardly at all, and (i) and (ii) did not occur the way we expected it. For (i), we assume that this is due to the fact that a third particle is needed in order to take up excess kinetic energy. If the kinetic energy is too high, the collisional cross section is too small and the partner cannot take up the excess, and the reaction partners move away from each other again, as described by Smith and Spanel, 2005. Interestingly, overall reagent ion counts of $NO^+$ and $O_2^+$ decrease at higher flow tube voltage, but the other ion counts do not increase at the same rate. We are unsure what causes this since we expected to see increased signals resulting from increased focusing of the ions. Perhaps they are hitting one of the accelerating electrode instead of being focused by the lenses, or their increased kinetic energy leads to a stronger deviation from the ideal ion path. In newer PTR-MS instruments, for example the PTR3 (Breitenlechner et al., 2017) and the Vocus PTR-ToF (Krechmer et al., 2018), such effects have been overcome by applying an additional focusing field. A similar modification could also be considered to further improve SIFT-MS sensitivity. We chose to use a flow tube voltage of 40 V as a compromise between increased water clustering and losing $NO^+$ and $O_2^+$ reagent ions.

Increasing the flow tube temperature also increases the kinetic energy, but randomly and for all molecules inside the flow tube, not just the ions. We expect effects of increased temperature to be similar to those for increased flow tube voltage, but instead found that the effects are rather small. One can see a slight decrease in product ion counts, cf. Fig. 4 (Fig. S5 for all ions) with increased flow tube temperature. However, reagent ion counts indicated a major shift due to decreases in interfering ions. Thus, a flow tube temperature as high as possible appears to be advantageous. However, the authors were concerned that for environmental samples, too high a temperature would reduce detection of more labile compounds with low thermal stability. Therefore, we decided on a flow tube temperature of 140°C.

Increasing the carrier gas flow while keeping the sample gas flow stable meant increasing the pressure in the flow tube. This both decreases the main free path of the ions and provides more collision partners. While reactions are expected to be more efficient since surplus energy can be dissipated more easily the same dissipation will form ions with less average energy

(closer to thermal equilibrium). The colder ions should be easier to focus leading to less diffusive loss to the flow tube walls, but the increased number of collisions should deflect ions stronger. We observe an optimum for the reagent ion counts (approx. 200 ccm for H3O+ and NO+, 50 ccm for O2+) with strong intensity decreases afterwards. Product ion counts increase with higher carrier gas flows, but interestingly, the ratios do not seem to change strongly (cf. Fig. 5, S 3 and S 4).

The decreasing reagent ion counts and increasing product ion counts are probably a result of the increased reaction efficiency. The increase in reagent ion counts is probably due to a minimized diffusive loss up to a certain point, before the increased reaction efficiency and increased ion deflection by collisions outweigh this effect. It is interesting that we do not see changes in adduct formation patterns and fragmentation patterns, as we expected adducts to be destabilized, and fragmentations to be pushed towards the most stable ions by the increased number of collisions. The behavior of methanol

was unexpected as well: Its counts are highest for low carrier gas flows which is counteracting the trends of the other product ions. What causes this is unclear, but might be due to a contamination in the system, as a similar effect is observed for sample gas flows, *cf.* Fig. 2. For a carrier gas flow, 312 ccm (4 TorrLs$^{-1}$) were chosen to ensure high product ion counts whilst not losing too much reagent ion intensity.

    We also tested helium and nitrogen as carrier gases, by optimizing the operating conditions with this carrier gas (Fig. S3-

S10) and calibration (Table S2-S4) at the optimized values. We observed higher sensitivity using nitrogen carrier gas, but also higher LODs and lower SNRs at 1 ppb. Further, humidity-sensitivity of the reagent ions was also higher with nitrogen carrier gas, as was instrument background. In both cases 6.0 quality gases were used and the nitrogen was even further purified with a filter, so that total amount of impurities should be similar for both gases. We thus attribute the higher background we observed with nitrogen to the higher collisional cross-section of nitrogen molecules compared to helium

atoms, which might have caused a higher ionization efficiency of the impurities in the nitrogen and the instrument itself, basically increasing the visibility of the impurities by increasing the amount of ionized background analytes. We also attribute the higher sensitivity we observed with nitrogen to the higher ionization efficiency. Final running conditions for the SIFT-MS were: 40 V, 140°C, 158 ccm (2 TorrL s$^{-1}$) Helium, and 100 sccm sample.

## SIFT-MS robustness over time

The company advertises that the SIFT-MS instrument is stable very long-term, that you do not need to calibrate but can just use their daily validation routine for quality-assurance (Syft Technologies Ltd., 2019). To test this, we performed three calibrations at 60% humidity on one day (day 1), and repeated this one week later (day 8). Standard calibration curves for α-pinene are shown in Fig. 6. Results obtained on the two days were compared using F- and t-tests on the slopes and the intercepts of the calibration curves, performed separately for each reagent ion. The slopes were not heteroscedastic and

differences between the two days were not statistically significant (p = 95%), whereas the F-tests failed for the intercepts. The heterogeneous variance t-tests on the intercepts again did not show statistically significant differences between the intercepts of day 1 and day 8 (p = 95%). Thus, both calibrations were not significantly different between both days, so a calibration can be used for at least a week.

    In addition, we tested the variance in signal intensity of the 2 ppb calibration point with time. Here, we included the two

interacting factors to which day and to which calibration the raw data measurements belong. The Bartlett tests did not show heteroscedasticity, and the two-way ANOVA with the two interacting factors day and number of calibration only showed significant difference of the day for the ion 19 u / 75 u/ acrolein / $C_3H_5O^+\cdot H_2O$, i.e. the water cluster of acrolein. Perhaps the air humidity of the produced calibration standard varied enough to make it statistically significant. Thus, over the course of a week, the calibrations appear to be stable and can be used to calculate mixing ratios. We are not aware of a similar study

published, however, Ammann *et al.* (2004) showed loss of detector signal intensity over a period of two months during their field experiment with PTR-MS. When comparing the signal intensity measured during weekly validation of the instrument,

we observe the same trend, Fig. S12. A Neumann trend test was negative for the ions (p = 95%, n = 10), but the signal appears to be dropping, and the trends might become significant over a longer time period. Combining the two experiments, we conclude that the calibration is stable over the course of days to weeks.

To test the robustness of calibrations over longer time periods, we compared the calibrations performed in May, December and January. This time span included changes to the instrument such as the o-ring change and a detector crash. Considerable variation in LOD and sensitivity we observed (Fig. S13) indicating that calibrations need to be performed regularly, especially following repair or instrument maintenance. Syft tackles this problem by providing a daily automated validation procedure, but this only validates at 2 ppm of the standard substances in dry air. We adjusted the procedure to our low mixing ratio regime by diluting the standard to 20 ppb for validation. For routine measurements of a rather stable system with rather high mixing ratios (above ppb level), *e.g.* clean room air quality monitoring, the standard procedure using a multiple-gas standard and daily validation with the adapted Syft routine should be sufficient. However, for accurate quantification of dilute analytes in a varying system with different humidities, for example our soil VOC emission monitoring during dryout-incubations from flooded to dry soil, we recommend calibrating the instrument before every experiment series.

**Humidity dependence of product ion intensities of the SIFT-MS**

Humidity can have a large influence on the product ion intensity when $H_3O^+$ reagent ions are used. For example, α-pinene at 10 ppb loses approximately one fourth of the ($H_3O^+$) product ion intensity, whereas the product ion intensity upon reaction with $NO^+$ or $O_2^+$ remains stable, see Fig. 7. Even for $H_3O^+$ ions, influences are mixed – for lower mass molecules like methanol (Fig. S13) and lower mixing ratios (α-pinene, Fig. 7), the effect appears to be less prominent.

For methanol, the intensity decrease of m/z = 33 u matches the intensity increase of m/z = 51 u, the water cluster (Fig. S13). Both are ca. 50 cps for the humidity increase from 30 % to 90 %. This could reflect either an increased association of water to protonated methanol in a three-body association involving a third collision partner M that takes up excess energy ($CH_3OH \cdot H^+ + H_2O + M \rightarrow CH_3OH \cdot H^+ \cdot H_2O + M^*$), or an increased ionization of methanol by $H_3O^+ \cdot H_2O$, where one water ligand is exchanged for methanol ( $CH_3OH + H_3O^+ \cdot H_2O \rightarrow CH_3OH \cdot H_3O^+ + H_2O$ ). $CH_3OH \cdot H^+ \cdot 2\, H_2O$ ($m/z(H_3O^+)$ = 69 u) could not be observed directly, as we used a mixed VOC standard and at this m/z, isoprene is also detected. A quick calculation of the isoprene signal we should see based on the isoprene signal we see at $m/z(NO^+)$ = 68 u showed us that most of the observed signal should be from isoprene and if at all only a minor amount of the methanol dihydrate ion should be present. For the exact calculation, please refer to the Supporting Information, S4.1.

However, for acetaldehyde, we do not see the same effect – a decrease of ca. 250 cps from 30 % to 90 % humidity is accompanied by an increase of ca. 50 cps of the water cluster (Fig. S14). This difference can also not be explained if one assumes that the protonated product is the product of the reaction with $H_3O^+$ and the water cluster is the product of the reaction with $H_3O^+ \cdot H_2O$ – the reaction rate difference is rather insignificant ($3.7 \cdot 10^{-9}$ vs. $3.1 \cdot 10^{-9}$ cm³ molecule$^{-1}$ s$^{-1}$).

Overall, for acetaldehyde, acetonitrile, and ethanol, the water cluster intensity rise does not match the intensity of decline of the primary product ion, whereas for methanol and acrolein, it does. Thus, high moisture sensitivity of the compound appears to correspond to this mismatch, whereas a low moisture sensitivity avoids it. In accordance with Wilson *et al.* (2003), we conclude that a back-reaction of the product ion with water occurs via a ligand exchange of $H_3O^+ \cdot M$: The analyte M is exchanged by water again and thus not part of the ion anymore, leaving a thermally colder reagent ion behind: $e.\,g.\, CH_3CHO \cdot H_3O^+ + H_2O \rightarrow H_3O^+ \cdot H_2O + CH_3CHO$ (Spanel and Smith, 1998). This affinity to $H_3O^+$ should correspond to the proton affinity of the compound, as $H_3O^+$ is essentially a proton with one water ligand associated. Kebarle *et al.* (1976), published proton affinities of 187.3, 196.8, and 185.4, and 182.3 kcal mol$^{-1}$ for acetaldehyde, ethanol, acetonitrile, and methanol. The difference was greatest for ethanol, having the highest proton affinity, and smallest for methanol. Only

acrolein does not fit to this picture as it has a proton affinity of 190.4 kcal mol$^{-1}$ (Del Bene, 1978), but since this value is from a different source, it might have been calculated differently.

We further evaluated the effect of humidity when normalizing the product ion counts to the reagent ion counts. Fig. 8 and S15 show that whether the absolute signal is humidity sensitive or not, both cases show a linear humidity dependence after being normalized. For all gases we tested, linear humidity dependence was observed for calibrations performed between 30-90% relative humidity, when product ion counts were normalized to reagent counts. Usually, the dry samples were in line with the other results as well. This is not the case for toluene at lower mixing ratios: in dry air, the relative intensity is lower than for humid samples, but for the humidified samples, the trend is the same as for higher mixing ratios. This might be caused by problems with the bypass line of the humidifiers.

**Evaluation of calibration procedures**

To account for humidity effects on ion counts, several calibration procedures were tested. When using the chosen settings, the humidity has to be taken into account.

For the humidity sensitive ions, we first investigated whether the humidity is better represented by the actual relative humidity or the ratio of the water cluster intensities, $\frac{I(H_3O^+ \cdot H_2O)}{I(H_3O^+)}$. Since the ratio of the intensities correlates quite linearly with the relative humidity (Fig. S16) and is easy to measure *in situ*, the representation of the humidity as the intensity ratio appears to be more useful. Second, we tried normalizing to both $I(H_3O^+)$ and $I(H_3O^+) + I(H_3O^+ \cdot H_2O)$. Normalizing to both reagent ions makes the ion count more humidity dependent, but it also appears to make the humidity dependence more linear and decrease the variance in the data, (Fig. S17). Thus, we decided to normalize to both reagent ions. One has to keep in mind though that this is only valid if they react with the analyte on a similar rate. If the kinetic rate constants are too different, the influence of the two reacting ions is not equal, so they should be treated differently. This is also why higher water clusters were not considered – they generally react roughly 1000 times slower.

To account for air humidity in the calibration, we tested the different methods described in the experimental section. Binning experimental results into humidity categories of 0 %, 30 %, 60 %, and 90 % as proposed in Eq. (2) and (3) are very uncertain when applied to intermediate humidity (e.g. 45%) where both calibration curves are not very close. Assuming a linear humidity dependence as in Eq. (4) and (5) does not necessarily reflect the trends observed for lower mixing ratios, *e.g.* Fig. S15 where responses are not as linear. In addition, a correction of the mixing ratio the instrument calculates was tested. This should be done carefully, as the results of all three reagent ions are averaged by the instrument if they do not differ too strongly, so one might actually induce error by correcting for humidity when the analyte is measured by multiple reagent ions. The most exact version is calibration function Eq (7), which is derived from the function Syft uses to calculate mixing ratios based on the instrument parameters, Eq. (17):

$$\chi = k_B \cdot \frac{T_{FT}}{P_{FT}} \cdot \left( \frac{\varphi_{carr}}{\varphi_{samp}} + 1 \right) \cdot \frac{\Sigma_{i=1}^{N}\left(I_{P_i} \cdot ICF_{P_i}\right)}{t_r \cdot br_i \cdot \Sigma_{j=1}^{M}\left(k_j \cdot I_{R_j} \cdot ICF_{R_j}\right)} \tag{17}$$

where $k_B$ is the Boltzmann constant, $T_{FT}$ is the flow tube temperature, $P_{FT}$ the flow tube pressure, $\varphi_{carr}$ and $\varphi_{samp}$ the carrier gas and sample gas flows, $I_{P_i}$ and $I_{R_j}$ are product and reagent ion counts, $ICF$ experimentally determined is the instrument calibration factor accounting for ion discrimination of each ion, $t_r$ is the reaction time, $br_i$ the branching ratio of the ion, and $k_j$ the rate constant of the reaction of the reagent ion with the analyte to form the product ion.

However, Eq. (8) quickly increases the number of parameters that need to be fitted: For example for methanol, the equation would be

$$\chi(ppbv) = a \cdot \frac{I(CH_3OH_2^+) + b \cdot I(CH_3OH_2^+ \cdot H_2O)}{I(H_3O^+) + c \cdot I(H_3O^+ \cdot H_2O)} + d \tag{18}$$

This is a four dimensional problem with four parameters. It cannot easily be plotted in two dimensions to see the quality of the fit, so one has to rely on the results of the fit without checking it visually. Using the ICFs determined during the validation reduces the number of fitted parameters, but still not the number of dimensions. The most versatile method we found is Eq. (9), deriving from Eq. (7) by multiplying the fraction by $\frac{1}{I(H_3O^+)} / \frac{1}{I(H_3O^+)}$. This way, the equation is reduced by one dimension, so that if there are no product ion water clusters, one can visualize the results in a 3D plot, see Fig. 9. As expected, the equation fits the data with a minimum of physical parameters without relying on experimentally determined parameters other than the ion intensities.

To compare the models, the Bayesian Information Criterion (BIC) was calculated for the calibration of each substance by each method, cf. Table S6. Although the BIC punishes for a bigger number of parameters, still, Eq.s (7) and (9), the calibration functions based on actual theoretical considerations, consistently have the smallest BIC and thus fit the data best. This fits the considerations above. Thus, for humidity dependent ions, this is the method of choice.

**Comparison of SIFT-MS to PTR-MS**

The optimized SIFT-MS was compared to our PTR-QMS 500 using the diluted Ionicon calibration standard (mixing rations between 0.25 and 10 ppb), and for each mixing ratio at 10 %, 30 %, 60 %, and 90 % relative humidity (25°C). Since to the knowledge of the authors parameters that access the quality of a calibration like LOD, sensitivity, SNR, precision and robustness are only established for a 2D calibration curve, for the following comparison with the PTR-MS, we used the simple humidity independent regression based on normalized ion counts, Eq. (3). This is the most accessible and the easiest to compare with the PTR-MS, especially because the humidity is known and does not need to be compared by $I(H_3O^+ \cdot H_2O) / I(H_3O^+)$. In the graphs, the results for 30 % humidity are shown, and the results for all humidities are summarized in Table S7-S12. The PTR-MS was operated at E/N = 136 Td with an inlet line temperature of 85°C to reduce water clusters of the product ions and possible condensation of water droplets in the tube. The authors are aware that this increases fragmentation reactions, however, we found the settings to work well for humid samples: The formation of m/z = 37 u and water clusters of product ions is reduced substantially. Also, we reduced the risk of water and VOCs condensing in the inlet tubes by using the stated high inlet temperature and drift tube temperature.

At all humidities, the limit of detection (LOD) was lower for our PTR-MS. While the LOD of the PTR-MS is between 10 and 100 ppt for most masses, the LODs of the SIFT-MS are generally one order of magnitude higher, between 100 ppt and 1 ppb, see Fig. 10 and Table S7–S8. For PTR-QMS-systems, this matches the LODs reported for other instruments as well (Yuan et al., 2017). This is probably due to three factors: First, the flow into the PTR-MS is about three times as high, so that more analyte is ionized. Second, the reagent ion counts (*e.g.* $H_3O^+$) are twice as high for the PTR-MS, doubling the number of product ions, thus more are detected. This was also discussed by Smith *et al.* (2014), who also report lower LODs for PTR-MS in their review. Third, the variation in the signal over time is much lower for PTR-MS, maybe due to more stable conditions and longer reaction times (approx. 5 ms in SIFT-MS vs. 0.2 – 1 s in PTR-MS, de Gouw, 2007) in the flow tube. This can also be inferred from the calibration curves of Lourenço, C. *et al.* (2017), as their R² value is lower for SIFT-MS than for PTR-MS. However, the difference in LOD between the instruments is smaller than was previously found: Blake *et al.*, 2009, estimated a difference of two orders of magnitude whereas we found only one order of magnitude. On the other hand, Milligan *et al.* (2007) presented a SIFT-MS with LODs in the mid-ppt-range, so very low values are possible on custom-built instruments, and for PTR-MS, the PTR-Qi-TOF (Sulzer et al., 2014) and the Vocus PTR-TOF (Krechmer et al., 2018) even having LODs reported below ppt for 1 s scan time. Still, instrument improvements by Syft over the last 10 years as well as our improvements to the SIFT-MS instrument significantly improved LOD.

For the sensitivity analysis, the slopes of the calibration curve based on ion intensities normalized to $10^6$ reagent ions were compared. In general, the SIFT-MS is more sensitive than the tested PTR-MS, and appears to become even more sensitive the higher the $m/z$ ratio becomes: For methanol, both instruments are comparable, for toluene, the sensitivity is at least twice as high, see Fig. S18 and Table S9-S10. These results are different from the results of Lourenço, C. *et al.* (2017), where PTR-MS shows a higher sensitivity by a factor of 10 and even higher sensitivities have been reported for the most recent PTR-MS developments (Sulzer et al, 2014, Breitenlechner et al., 2017, Krechmer et al., 2018), but match the reports of Prince *et al.* (2010) for SIFT-MS sensitivity. Still, due to a higher precision of the PTR-MS data that is also reflected in the much lower LOD, the signal to noise ratio at 1 ppb is still much higher for the PTR-MS than for the SIFT-MS.

This also influences the signal to noise ratios, see Fig. S19 and Table S11-S12. For smaller masses, the tested PTR-MS has a much higher SNR, whereas for the higher masses of the aromatic molecules like dichlorobenzene, *o*-xylene and toluene, the SIFT-MS has a higher SNR. With these molecules, the tested PTR-MS has a comparable LOD and a lower sensitivity, so this adds up to a lower SNR. However again, with the higher sensitivity and lower LODs mentioned in the literature (Yuan et al., 2017), a higher signal to noise ratio should be found on state-of-the-art PTR-TOF-MS instruments. Still, the SIFT-MS has the advantage that isomeric compounds can be separated by the different reactions with different reagent ions, so for these analytical problems, it is better to use. Plus, the sensitivity is already low enough for regular atmospheric trace gas measurements, so it can be used as a robust lower-cost easy-to-use alternative to the PTR-MS.

**Conclusions**

We successfully improved a purchased SIFT-MS to meet the requirements of sub-ppb atmospheric trace gas measurements. Hardware improvements like changing o-rings in the purchased instrument for materials with lower degassing, and exchanging the capillary in the inlet system with a VICI valve helped reduce the SIFT-MS background. Increasing the sample gas flow by a factor of 5 also improved sensitivity greatly, but made adjustments of the carrier gas flow, the flow tube voltage and temperature necessary. In total, we achieved a decrease of the SIFT-MS' LOD by a factor of 10. The humidity-dependence resulting from the high sample gas flow could be corrected by a humidity-dependent calibration. The SIFT-MS is stable over shorter time periods, as we could demonstrate by comparing calibrations a week apart that are not significantly different. However, it shows considerable variations in signal intensity over longer periods, so that at least after each maintenance, the instrument should be calibrated: the LOD varied by up to a factor of two, the sensitivity by up to a factor of three. This drawback was addressed by Syft by implementing a workdaily validation routine that takes approx. 10 min that we adjusted to work for low mixing ratios, so the instrument calibration factor balancing out the mass discrimination should account for those instabilities. Still, we calibrate our instrument with humidity before every experiment series in addition to the one-point validation of the SIFT-MS procedure.

The comparison of SIFT-MS and PTR-MS confirmed that PTR-MS has a lower LOD than SIFT-MS, though modification of the SIFT-MS instrument improved its LOD to within an order of magnitude of the PTR-QMS. Both instruments are equally sensitive when responding to signal changes and have similar dynamic range. The calibration at multiple humidities demonstrated that PTR-QMS is more humidity dependent than SIFT-MS, indicating that it is important to calibrate for the humidity as well, or take care that it remains constant during measurements. However, the Vocus PTR-TOF has overcome the humidity-dependence by introducing high humidity in the drift tube (Krechmer et al., 2018). Still, the additional structural information that can be gained by SIFT-MS is especially helpful for mixtures of isomers like acetone and propanal. Overall, SIFT-MS is a good lower-cost alternative to PTR-MS for analyzing gases with a more complex mixtures of compounds including isomers at varying humidity.

**Code and data availability**

Code and data are published as Lehnert, A.-S., Behrendt, T., Ruecker, A., Pohnert, G., Trumbore, S. E.: Max Planck Society, https://dx.doi.org/10.17617/3.2u, 2019.

**Author contribution**

5    The study was designed by A.-S. L., T. B. and G. P. Experiments were performed by A.-S. L. with assistance from T. B. and A. R. Data processing and analysis was done by A.-S. L. with assistance from T. B. and A. R. Manuscript was written by A.-S. L. All authors assisted with data interpretation, discussion of results and helped to improve the quality of the manuscript.

**Competing interests**

The authors declare that they have no conflict of interest.

**Acknowledgements**

Thanks to Paul Wilson, Vaughan Langford, Mar Viñallonga and their colleagues at Syft Technologies Inc., New Zealand, for technical assistance during the instrument modifications as well as feedback to methodological considerations. We thank Marvin Shaw (University of York, GB) for input on how to decrease contamination by removing the carrier gas shut down valve. A.R. and A.-S. L. received financial support from the Deutsche Forschungsgemeinschaft (DFG) in the frame of the

collaborative research center CRC 1076 AquaDiva.

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

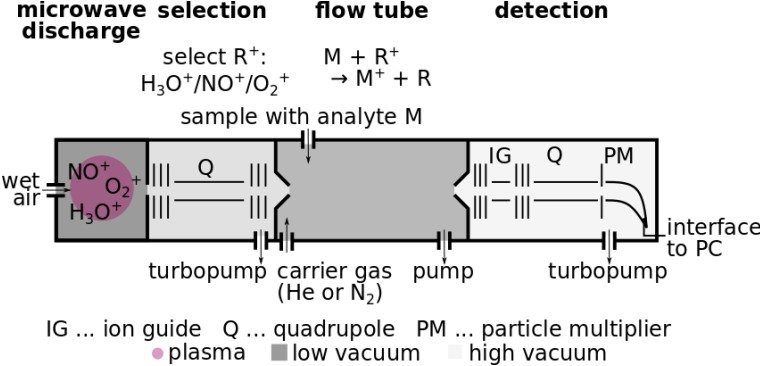

**Figure 1:** Scheme of SIFT-MS instrument components. SIFT-MS generates a plasma from wet air and then selects the reagent ion R ($H_3O^+$, $NO^+$ or $O_2^+$) via a quadrupole (Q). In the flow tube, reagent ions $R^+$ and analytes M meet and react. Their reaction time is defined by the flow of the carrier gas through the tube and a small electric field to focus the ions. The ions are detected *via* a quadrupole-MS.

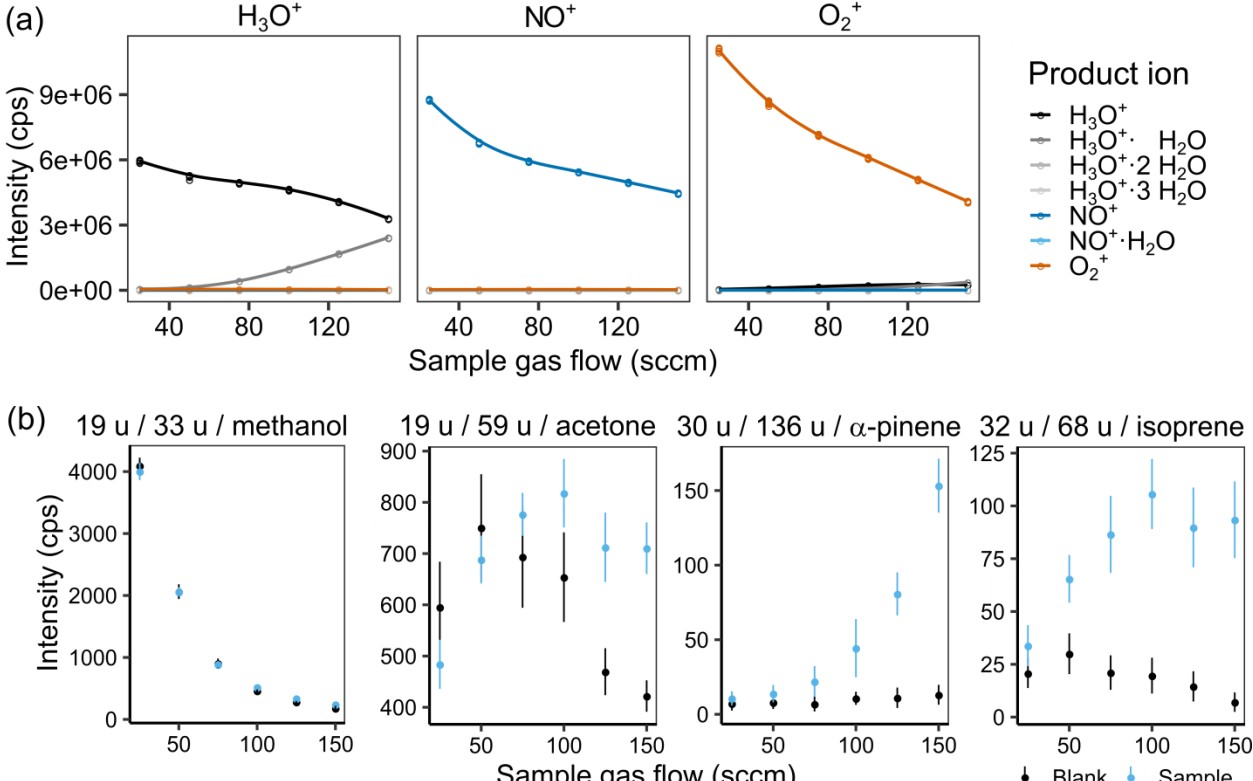

**Figure 2:** Effect of the sample gas flow on the intensity of the different measured product ions when selecting for $H_3O^+$ (left), $NO^+$ (middle) and $O_2^+$ (right) in the first quadrupole (a) and the product ion intensity for a Zero Air blank and a 1ppb VOC standard sample for methanol, acetone, α-pinene and isoprene at 60% humidity (25°C) (b). In (b), the captions are labelled with *m/z* of the reagent ion, *m/z* of the product ion, and the corresponding substance. The helium carrier gas flow was kept at 158 sccm (2 TorrL s[-1]).

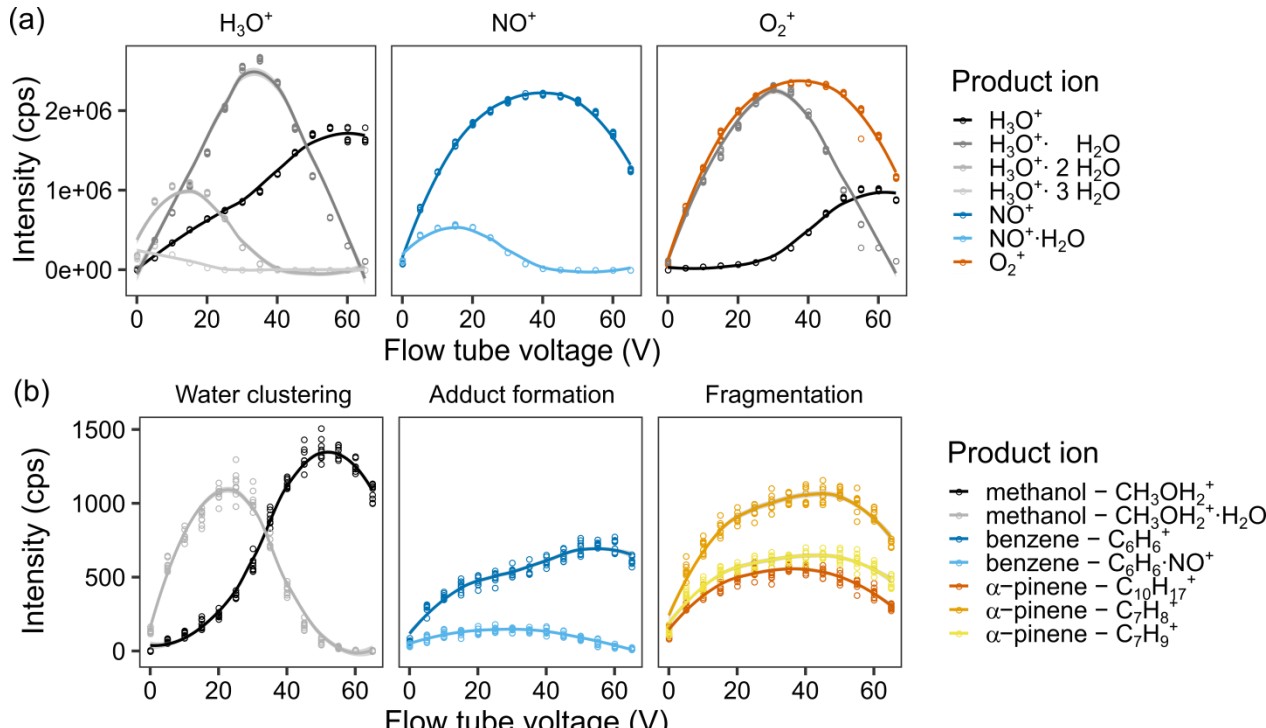

**Figure 3:** Effect of the flow tube voltage on reagent and product ion counts. (a) intensity of the different measured product ions when selecting for $H_3O^+$ (left), $NO^+$ (middle) and $O_2^+$ (right) in the first quadrupole. (b) examples of the product ion behaviour illustrating the effect of water clustering on the methanol ions reacting with $H_3O^+$ (left), adduct formation on the benzene ions upon reaction with $NO^+$ (middle), and fragmentation of the α-pinene ions upon reaction with $O_2^+$ (right). Measurements were done for a humid (90% at 25°C) 5 ppb VOC standard air flow, and were fit *via* LOESS. For the results of all ions, cf. Fig. S9.

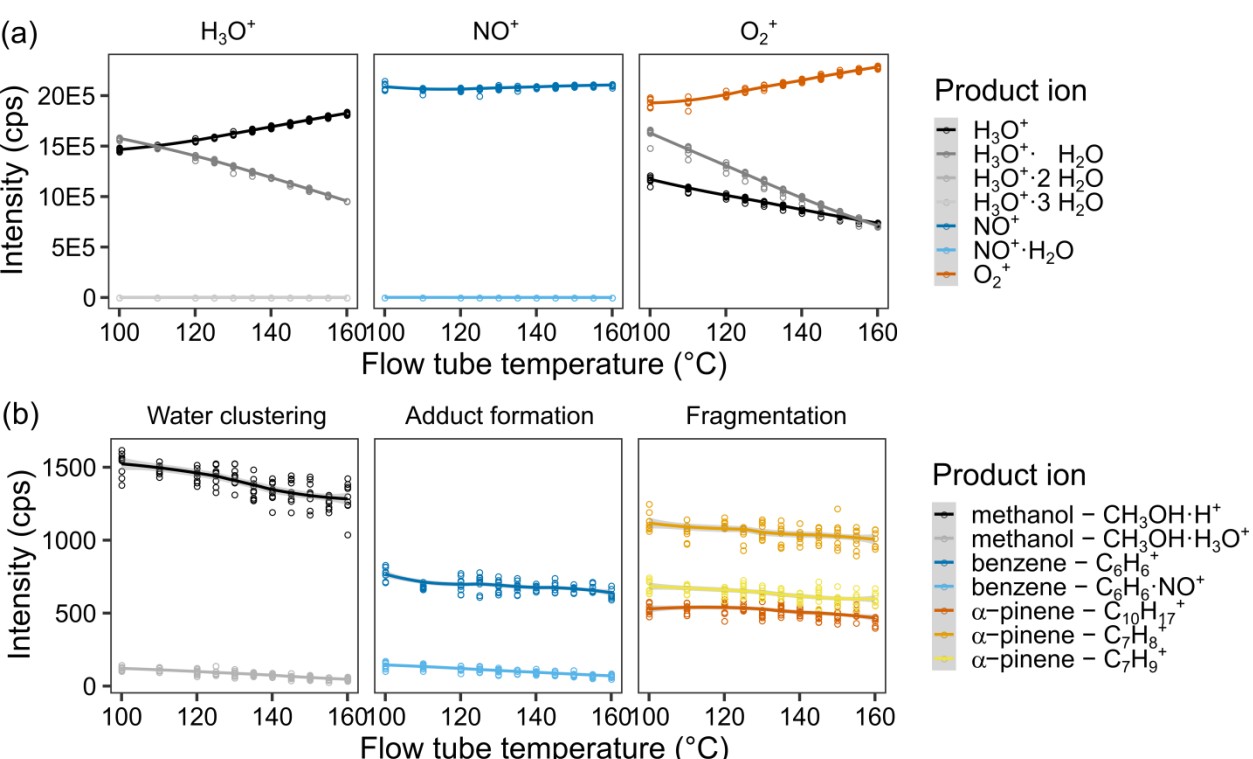

**Figure 4:** Effect of the flow tube temperature on reagent and product ion counts. (a) intensity of the different measured product ions when selecting for $H_3O^+$ (left), $NO^+$ (middle) and $O_2^+$ (right) in the first quadrupole. (b) examples of the product ion behaviour illustrating the effect of water clustering on the methanol ions reacting with $H_3O^+$(left), adduct formation on the benzene ions upon reaction with $NO^+$(middle), and fragmentation of the α-pinene ions upon reaction with $O_2^+$(right). Measurements were done for a humid (90% @ 25°C) 5 ppb VOC standard air flow, and were fit *via* LOESS. For results of all measured ions, cf. Fig. S8.

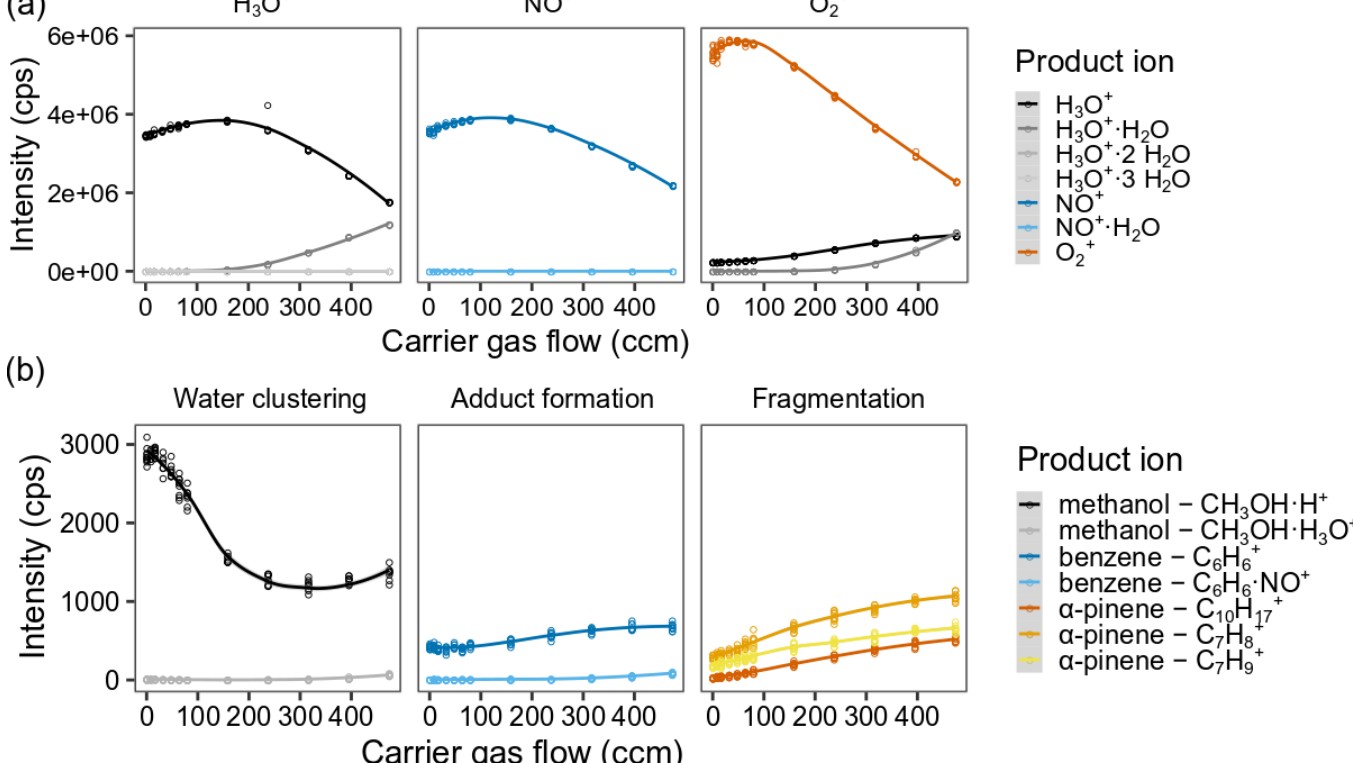

**Figure 5:** Effect of the helium carrier gas flow on reagent and product ion counts. (a) intensity of the different measured product ions when selecting for $H_3O^+$ (left), $NO^+$ (middle) and $O_2^+$ (right) in the first quadrupole. (b) examples of the product ion behaviour illustrating the effect of water clustering on the methanol ions reacting with $H_3O^+$ (left), adduct formation on the benzene ions upon reaction with $NO^+$ (middle), and fragmentation of the α-pinene ions upon reaction with $O_2^+$. Measurements were done for a humid (90% @ 25°C) 5 ppb VOC standard air flow, and were fit *via* LOESS. The sample gas flow was 120 sccm (capillary with 0.010" inner diameter). For complete results of all measured ions, cf. Fig. S7.

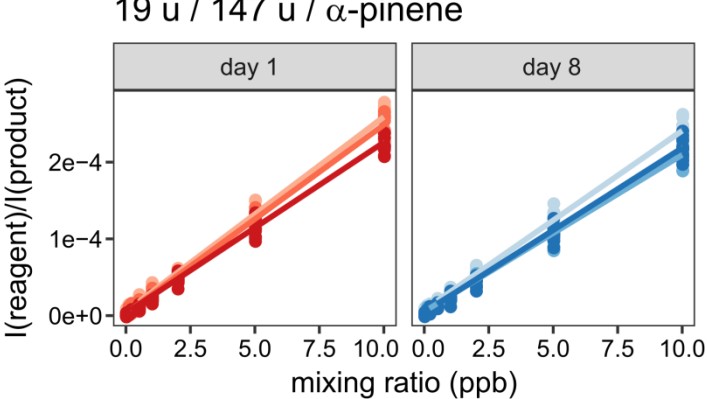

**Figure 6:** Robustness of the α-pinene calibration of the SIFT-MS. Three calibrations were conducted on one day, and one week later, on day 8. Slopes and intercepts were not significantly different (p = 0.9986, n = 3) between the days.

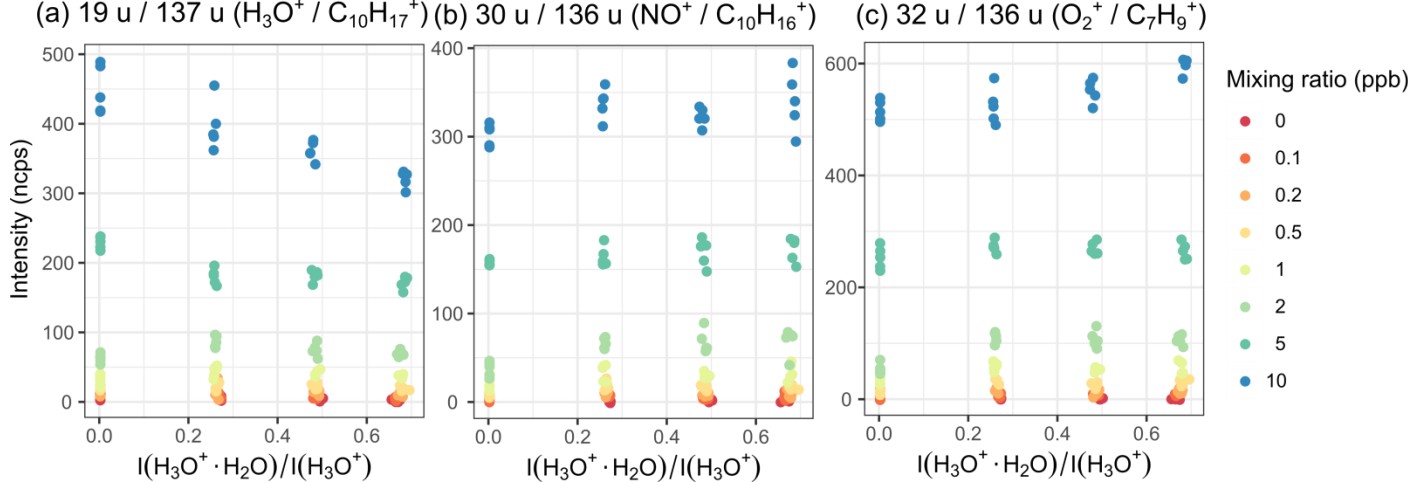

**Figure 7:** Humidity dependence of α-pinene signal at mixing ratios between 0.1 and 10 ppb upon reaction with the different reagent ions $H_3O^+$ (a), $NO^+$ (b), and $O_2^+$ (c) forming the product ions $C_{10}H_{17}^+$, $C_{10}H_{16}^+$, and $C_7H_9^+$. Humidity is measured as the ratio of the $H_3O^+ \cdot H_2O$ and the $H_3O^+$ intensity, cf. Fig. S20.

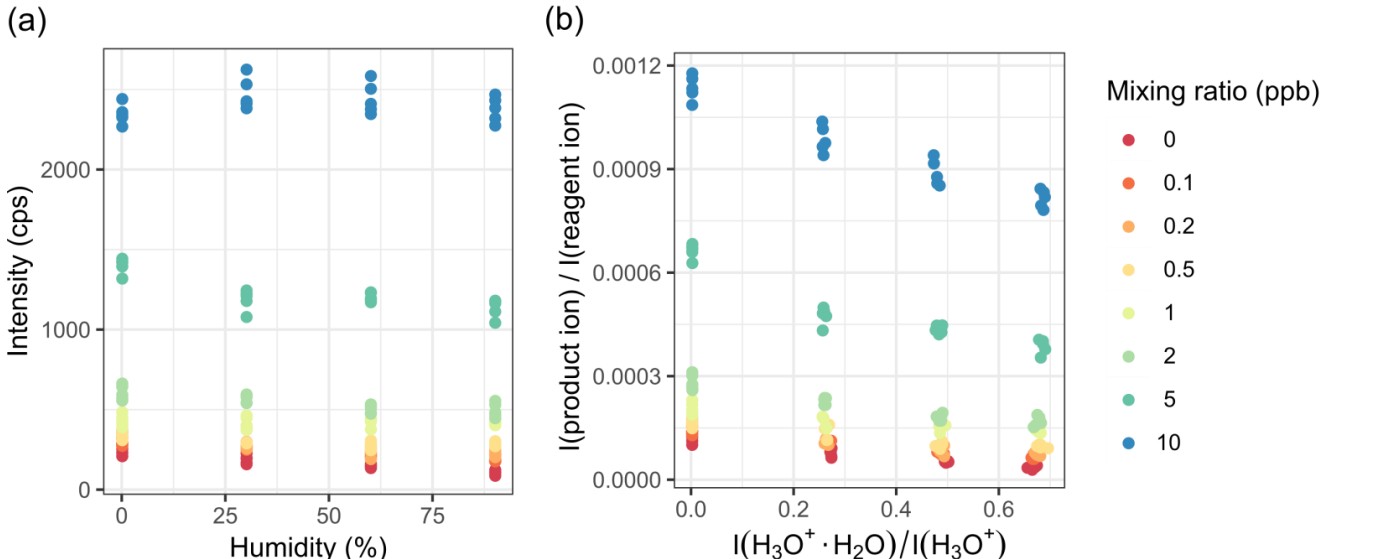

**Figure 8:** Humidity dependence of methanol m/z = 33 amu intensity. (a) Absolute counts vs. relative humidity at 25°C. (b) Relative intensity per reagent intensity vs. the ratio of $H_3O^+$ and its first water cluster as a measure of humidity.

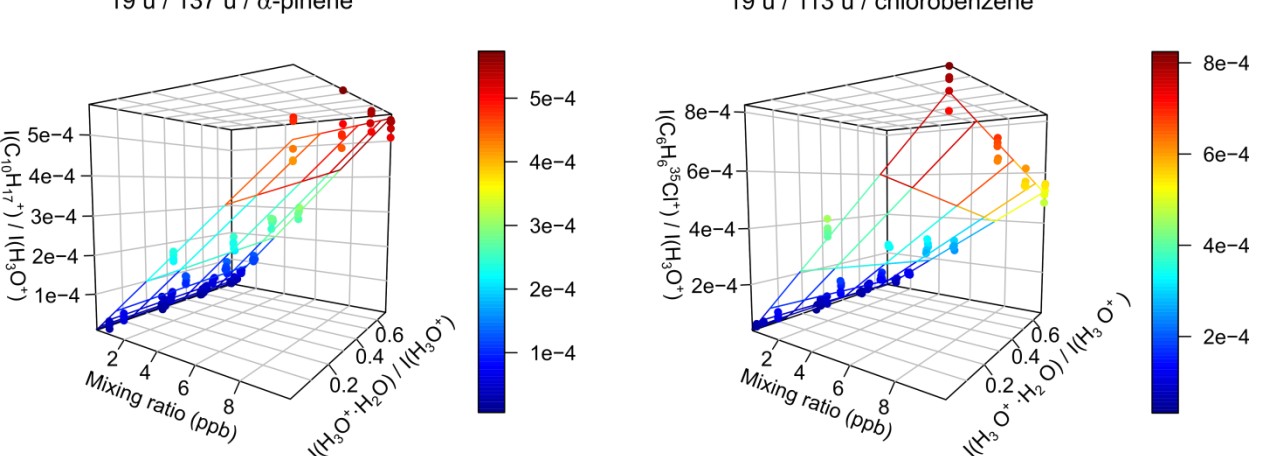

**Figure 9:** Relative product ion intensities and calibration plane for α-pinene and chlorobenzene using Eq. (9). Even strong humidity-dependence like for chlorobenzene is accounted for using this method. $I(C_{10}H_{17}^+)/I(H_3O^+)$ and $I(C_6H_6^{35}Cl^+)/I(H_3O^+)$ are the relative product ion intensities of the two mentioned ions, $I(H_3O^+ \cdot H_2O)/I(H_3O^+)$ serves as measure for the humidity.

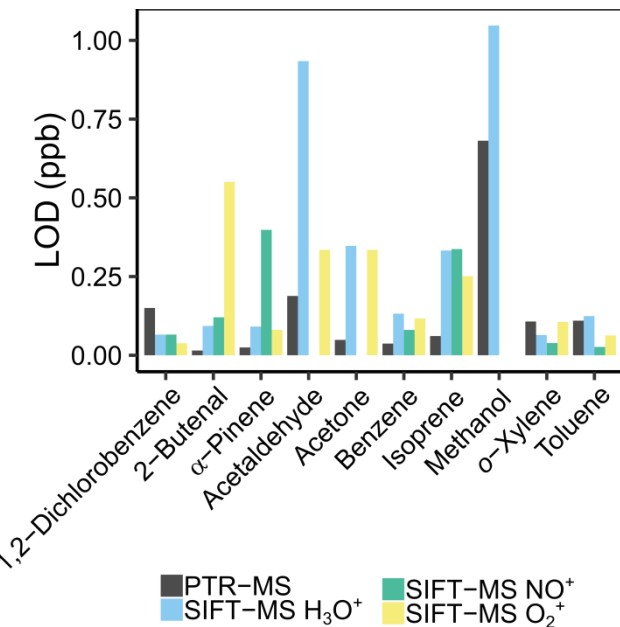

**Figure 10:** Comparison of the limit of detection (LOD) for PTR-MS and the different reagent ions of the SIFT-MS of the shown VOCs at 30 % humidity.