# Peer review of "SIFT-MS optimization for atmospheric trace gas measurements at varying humidityAnn-Sophie Lehnert1,2, Thomas Behrendt1, Alexander Ruecker1, Georg Pohnert2, Susan E. Trumbore1"

_Atmospheric Measurement Techniques, 2019_

## Referee Comment (RC1) · Anonymous Referee #1 · 10 Dec 2019

General comments:

This paper details a set of modifications to a commercial SIFT-MS instrument that improved the LODs, backgrounds, and sensitivities. The authors have also completed an expansive set of experiments characterizing the modified SIFT instrument's operational parameters and performance. While the work is thematically simple, I think rigorous instrument characterization papers like this are useful to those in the community who measure VOCs with SIFT or other atmospheric chemical ionization mass spectrometers. I recommend this work be published after significant contextual and written issues with the paper are addressed.

[Figure]

Particularly, the PTR comparison portion of the work is afterthought and has numerous problems. Instead, I suggest refocusing the paper around the modified SIFT instrument.

My issues with the PTR portion of this work are thus:

1) The authors portray a very old quadrupole PTR-MS as a generic PTR instrument representative of all PTR-MS instruments. There are many different PTR-MS instruments with different performance levels. Importantly, almost all of them have higher performance characteristics than the PTR-QMS 500, which has not been state-of-the-art for a long time.

Lately, newer models such as the PTR Qi-TOF (Sulzer et al. 2014), PTR6000 X2, PTR3 (Breitenlechner et al.), and Vocus (Krechmer, Lopez-Hilfiker, et al.) have much higher sensitivity and much lower LODs. The PTR-MS sensitivities in Figure 4 are poor relative to those of the PTR3, which have been demonstrated to be three orders of magnitude larger. And very relevant to this work, the Vocus PTR-MS has been shown to have no quantitative humidity dependence, making the broad statement on P10 L39 no longer true.

The PTR-MS field has largely moved to time-of-flight mass analyzers that offer much higher mass resolution (as opposed to the quadrupole demonstrated here), fast full-spectrum acquisitions, high sensitivity, and low backgrounds.

While the authors are fair and unbiased in their comparisons here, I fear that measuring a highly modified SIFT against a much older PTR is not representative, nor does it offer useful information to the field. As they acknowledge, SIFT/PTR intercomparisons have been published before for other applications. I do not think there is enough different about the proposed application here that warrants another intercomparison.

2) There is little-to-no mention of "current" PTR literature or instrumental developments to put this paper in context. For example, the improvements made to the SIFT instrument in this work seem to be similar in spirit to those made to the PTR QMS in Deming et al. 2018. Indeed, additional information about the SIFT-MS's time dependence from an experiment similar to Deming et al. would be useful in this paper.

Separately, I believe the authors left far too much information in the SI. Many of the SI plots are important to understanding the SIFT's performance characteristics and are referenced several times. They also clearly involved a lot of work and would be useful to many users. I think it would be a shame if they remained hidden. I would suggest cleaning some of them up and moving them back to the main paper. There is, after all, no length limit in AMT.

In summary, I recommend that this paper be published but suggest that the authors reframe the narrative of the paper around improving and characterizing the SIFT instrument, deemphasize the comparison with the PTR instrument, and bring a large portion of the impressive SI work into the main paper.

Specific comments:

Section 3.1 seems to repeat a large amount of the material in Section 2.2. A description of the SIFT improvements should be in one or the other, but not both.

P3 L7: "...usually a quadrupole MS". Do the authors have evidence of this? For SIFT this is true, but TOF has been the dominant technique for PTR for almost a decade now. This is not trivial, as the issues with how the authors characterize PTR are serious throughout the work.

P4 L16-18: The authors judge the success of their instrument modifications here by evaluating the background. Did they also evaluate response time, which is a critical parameter for ambient atmospheric measurements? PEEK may have a higher background, but its response time has been shown to be significantly better than stainless steel (coated and non). (Deming et al., 2018)

P10 L17: Mass-to-charge ratios should have a label as such everywhere in the paper.

e.g. m/z 19 or m/Q 75

Technical corrections: P4 L33: This is the methods section. Are the authors referring to a different methods section? P5 L20: should be "it more strongly punishes a larger number of parameters" P6 L25: "workdaily" I'm not familiar with this word.

References:

Breitenlechner, M., Fischer, L., Hainer, M., Heinritzi, M., Curtius, J. and Hansel, A.: PTR3: An Instrument for Studying the Lifecycle of Reactive Organic Carbon in the Atmosphere, Anal. Chem., 89(11), 5824–5831, doi:10.1021/acs.analchem.6b05110, 2017.

Deming, B. L., Pagonis, D., Liu, X., Day, D. A., Talukdar, R., Krechmer, J. E., de Gouw, J. A., Jimenez, J. L. and Ziemann, P. J.: Measurements of delays of gas-phase compounds in a wide variety of tubing materials due to gas–wall interactions, Atmos. Meas. Tech., 12(6), 3453–3461, doi:10.5194/amt-12-3453-2019, 2019.

Krechmer, J., Lopez-Hilfiker, F., Koss, A., Hutterli, M., Stoermer, C., Deming, B., Kimmel, J., Warneke, C., Holzinger, R., Jayne, J., Worsnop, D., Fuhrer, K., Gonin, M. and De Gouw, J.: Evaluation of a New Reagent-Ion Source and Focusing Ion–Molecule Reactor for Use in Proton-Transfer-Reaction Mass Spectrometry, Anal. Chem., 90(20), 12011–12018, doi:10.1021/acs.analchem.8b02641, 2018.

Sulzer, P., Hartungen, E., Hanel, G., Feil, S., Winkler, K., Mutschlechner, P., Haidacher, S., Schottkowsky, R., Gunsch, D., Seehauser, H., Striednig, M., Jürschik, S., Breiev, K., Lanza, M., Herbig, J., Märk, L., Märk, T. D. and Jordan, A.: A Proton Transfer Reaction-Quadrupole interface Time-Of-Flight Mass Spectrometer (PTR-QiTOF): High speed due to extreme sensitivity, Int. J. Mass Spectrom., 368, 1–5, doi:10.1016/j.ijms.2014.05.004, 2014.

---

## Referee Comment (RC2) · Patrik Spanel (Referee) · 28 Apr 2020

The paper presents results of extensive an thorough comparison of the current SIFT-MS instrument Voice 200 ultra with an older version of PTR-MS: QMS 500. Whilst the data are interesting, the results must not be interpreted as a test showing advantages and disadvantages of the current products on the market. The results are nevertheless based on honest and independent experimental work and should be published, as they do represent important contribution to validation of SIFT-MS in particular for VOC emissions from soils at different humidity.

It is worthy of note that PTR-MS is used at 136 Td, some clear justification should be

given for this value. Perhaps a lower value would reduce fragmentaion?

The important realization is, that the background to a large degree originates from impurities in the instrument components.

A very interesting finding relates to comparatively novel use of N2 carrier: "Nitrogen carrier gas lead to a higher sensitivity, but worse LODs and SNRs at 1 ppb and showed a higher humidity-sensitivity of the reagent ions, so we decided to use helium."

Clearly more research needs to be done on the N2 carrier gas.

"Final running conditions for the SIFT-MS were: 40 V, 140°C, 158 sccm Helium, and 100 sccm sample." This is a comparatively large proportion of sample, presumably following the work of Marvin Shaw, as indicated in section 2.2.

Does the worse LOD for nitrogen mean a higher backround? If so, is it the purity of the nitrogen that is the problem?

The ion chemistry discussed in 3.2 would perhaps deserve better terminology and insight instead of "intensity loss" of m/z 33 and gain of 51 it would be better to mention, for example, occurence of three body association of protonated methanol with water. Was m/z 69 present at all for methanol at 140C?

The discussion on acetaldehyde and proton affinity would also do with use of established terminology. The term "deionize" does not seem to be the best choice. What is likely to happen is a sequence of association reactions followed by final ligand switching to form H3O+(H2O)n

MH+H2O(n) + H2O -> H3O+H2O(n) + M

So the charge moves back into the reagent ion system. See for example our early paper on formaldehyde Spanel et al. RCM 13, 1354 (1999) or on hydrocarbons, equation 7 in Spanel & Smith IJMS 181 (1998) 1.

"In accordance with Wilson et al. (2003), we conclude that a back reaction of the

product ion with water might deionize the product ion to form a thermally colder reagent ion again and that this might correspond to the proton affinity of the compound"

This is more to do with apparent H3O+ affinity than H+ affinity.

I agree with the other referee in that much of the data could be in the text and not in Suppl.

―――――――――――――――――

---

## Referee Comment (RC3) · Anonymous Referee #3 · 8 May 2020

This paper provides some interesting original research into the detection of adulteration of Olive Oils. It should be published but a significant revision is needed before this can be achieved as there are a number of issues that require clarification.

I would ask the author to consider the following points:

1) The entire paper needs a revision by a fluent English speaker. There are too many changes that are required to mention but I have added a few in my comments.

2) There should be one place or alternatively a first mention in the paper where the following abbreviations are defined: PLSR (Partial Least Squares Regression Introduction and P2, L47); SIM (Selected Ion Mode) P5 L175; SPSS L132, P6; LOX (?

Lipoxygense, P7 L164); ICD (Interclass Distances (P7, L175). Some of the abbreviations are defined by the author in the text eg SECV P8, L192 but it might be simpler to have a single list of abbreviations together.

3) On P5, a description of the technique and methodology is outlined and a list of rate coefficients used for the analyses is given in Table 2. It is important to state what the bath gas used for the measurements is, as many of the concentrations of volatiles are derived from NO+ which undergoes association reactions. The rate coefficients for association are very different for a He bath gas than for a N2 bath gas.

4) Is a reference required in L130 for Tukey?

5) On P6, Line 151 with reference to Table 2. A more detailed description in the Table footnote as to the meaning of the letters would help.

6) The statistical results summarised in Table 4 need further explanation.

7) Figure 2 did not print.

8) Fig 3 is distorted and needs attention.

Some possible improvements to the written explanations and accounts follow:

Highlights

SIFT-MS can be used to detect adulteration in EVCOO as a rapid and simple analytical technique P2 L35 oils. . .

L37 . . .EVOO has become. . . . . ..adulterated foods. . . The most widely used. . .

L43 . . .study focused on the measurement of volatile composition..

L48 adulteration levels were clearly evident in most samples.

P3L62 . . .. . ..information sources dating back to the ancient Romans. . ...related to olive oil

L64-68 ....may not necessarily result in a food security problem, but it does result in substitution of a more valuable product with a cheaper and lower quality option. When this substitution occurs, the food industry might face decreased market acceptance, increased costs due to.... damaged brand and or bankruptcy amounting to billions...

P3 L62 .....information sources dating back to ancient Romans... incidents related to olive oil...

L64 ... Adulteration may not necessarily result in a food security problem, but it does result in substitution of a more valuable product with a cheaper and lower quality option.

L66-68 ....When this substitution occurs, the food industry might face a decreased market acceptance, increased costs due to recall, .... damaged brand and or bankruptcy amounting to billions of dollars...

L76 ...Frankel,2010). However, the presence of minor components in olive oil have an important role to distinguish well-known sensorial characteristics (Morales...

P4 L83 ... of them include isolation.....

L90 ....real-time monitoring....

L102 Several extra virgin......(EVOO) samples were collected...

L106 The oils were stored in the dark at room temperature until the analysis was undertaken

P5 L109 A 5g .... Was transferred to a 500 mL

P6 L136 .... Oil samples depending on their...

L135 ethanol is mentioned twice

P8 L181 profile of EVOOs, adulterated EVOO samples had ICDs that varied. When ....

P9 L209 has led to the application of new techniques to detect adulteration of olive oil

by cheaper additives. In this study, SIFT-MS has been proposed. . .

1. Is the subject of the paper consistent with the scope of the journal? Y Yes

2. As far as you know, has the material been published before in English? N No

3. Does the scientific content of the paper justify the space it will occupy? Yes

4. Can any parts of the paper be shortened or omitted without loss of scientific content? N

5. Are there any errors of fact or logic? See comments in letter to author.

6. Are all the figures necessary? Y

7. Is the number of significant figures in the tables justified as far as you know? Y

8. If the paper contains graphs and tables based on the same data, are both necessary? Y

9. Does the summary (normally about 50-100 words) bring out the main points of the paper? Y

10. Is the title suitable and adequate? Y

11. Are the literature references adequate? Y

12. Type of article. Please mark one of the below using x or *

\_\_\_ Review

\_\_\_ Short review

\_\_\_ Paper

\_x\_\_ Short paper

\_\_\_ Comment
___ Note

___ Other

13. What is your overall recommendation? Please mark one of the below using x or *

___ Publish as submit

_*_ Publish with major revision

__ Publish with minor corrections

___ Do not publish

---

## Author Comment (AC1) · 19 May 2020

**General response:**
We thank both Reviewers for their overall positive evaluations and the very helpful comments that helped to improve the quality of our manuscript. Our detailed responses to each comment made are provided below. As noted by both reviewers, the instruments used for measuring volatile organic gases are evolving rapidly, which means the actual results for the tests made in our manuscript may differ in newer instruments. However, we think the value of the work here is twofold: (1) these tests point to issues with current instrumentation that can help in further instrument

developments and use of existing instruments and (2) the types of tests made indicate issues that need to be addressed when evaluating instrument performance. To reflect these issues we have also changed the title of the overall manuscript, now "SIFT-MS optimization for atmospheric trace gas measurements at varying humidity." Both reviewers also requested that materials previously presented in the Supplemental Information (SI) be moved to the main text. We have done this, and this is responsible for most of the text changes in the revised document. Finally, we have gone through the text to remove imprecise language and improve grammar. We hope that the editor and reviewers will agree with the improvements to the revised manuscript.

**Response to Reviewer 1**

*Reviewer 1: 1) The authors portray a very old quadrupole PTR-MS as a generic PTR instrument representative of all PTR-MS instruments. There are many different PTR-MS instruments with different performance levels. Importantly, almost all of them have higher performance characteristics than the PTR-QMS 500, which has not been state-of-the art for a long time.Lately, newer models such as the PTR Qi-TOF (Sulzer et al. 2014), PTR6000 X2, PTR3 (Breitenlechner et al.), and Vocus (Krechmer, Lopez-Hilfiker, et al.) have much higher sensitivity and much lower LODs. The PTR-MS sensitivities in Figure 4 are poor relative to those of the PTR3, which have been demonstrated to be three orders of magnitude larger. And very relevant to this work, the Vocus PTR-MS has been shown to have no quantitative humidity dependence, making the broad statement on P10 L39 no longer true.The PTR-MS field has largely moved to time-of-flight mass analyzers that offer much higher mass resolution (as opposed to the quadrupole demonstrated here), fast full spectrum acquisitions, high sensitivity, and low backgrounds. While the authors are fair and unbiased in their comparisons here, I fear that measuring a highly modified SIFT against a much older PTR is not representative, nor does it offer useful information to the field. As*

*they acknowledge, SIFT/PTR intercomparisons have been published before for other applications. I do not think there is enough different about the proposed application here that warrants another intercomparison.*

The reviewer is correct, that PTR-TOF-MS is now the state-of-the-art instrument used by many researchers. However, the PTR-QMS which we used is still used in a number of laboratories and remains useful for general atmospheric measurements; we did not have a PTR-TOF-MS available for further comparisons. As noted above (and also be Reviewer 2), instruments evolve and the exact results – while important for evaluating the instruments we tested – will clearly differ as instruments evolve. What is important are the approaches we used for determining instrument performance. We do see the point made by Reviewer 1 that comparisons between an old instrument and a newer one might give a false impression about general performance. Therefore, we limited the direct comparisons likely to be instrument-dependent. Specifically, we removed the sections about humidity-dependence and the upper limit of the dynamic range, moving the sensitivity and SNR and humidity-dependence plot to the supporting information (now Fig. S18-S20). We also included references to literature values of newer PTR instruments and discussed the newer developments of PTR-MS in the introduction. We also made sure that it is clear we are talking about the specific PTR-MS instrument we tested. Examples:
P3 L6: All ions are then analysed by a mass spectrometer (MS), usually a quadrupole-MS for SIFT-MS and a time of flight-MS for PTR-MS, separating the ions by their m/z ratio and then counting the number of ions hitting the multiplier.
P11 L37: . . . and for PTR-MS, the PTR-Qi-TOF (Sulzer et al., 2014) and the Vocus PTR-TOF (Krechmer et al., 2018) even having LODs reported below ppt for 1 s scan time.
P12 L4: These results are different from the results of Lourenço, C. et al. (2017), where PTR-MS shows a higher sensitivity by a factor of 10 and even higher sensitivities have been reported for the most recent PTR-MS developments (Sulzer et al, 2014,

Breitenlechner et al., 2017, Krechmer et al., 2018), but match the reports of Prince et al. (2010) for SIFT-MS sensitivity.
P12 L22: However again, with the higher sensitivity and lower LODs mentioned in the literature (Yuan et al., 2017), a higher signal to noise ratio should be found on state-of-the-art PTR-TOF-MS instruments.
P12 L35: However, the Vocus PTR-TOF has overcome the humidity-dependence by having a high humidity in the drift tube (Krechmer et al., 2018).

*Reviewer 1: 2) There is little-to-no mention of "current" PTR literature or instrumental developments to put this paper in context. For example, the improvements made to the SIFT instrument in this work seem to be similar in spirit to those made to the PTR QMS in Deming et al. 2018. Indeed, additional information about the SIFT-MS's time dependence from an experiment similar to Deming et al. would be useful in this paper.*

We have included reference to the more recent PTR-MS literature, see above. With regards to the suggestion to include experiments similar to Deming et al., 2018: we usually do not observe delay times in the signal developments of more than 30 s. The inlet capillary is also much shorter than the other lines connecting the multi-port inlet to the actual flow-tube, so we do not think the effect is large. With the needle valve, we could also shorten the length of silcosteel lines from the inlet to the detector.

*Reviewer 1: Separately, I believe the authors left far too much information in the SI. Many of the SI plots are important to understanding the SIFT's performance characteristics and are referenced several times. They also clearly involved a lot of work and would be useful to many users. I think it would be a shame if they remained hidden. I would suggest cleaning some of them up and moving them back to the main paper. There is, after all, no length limit in AMT. In summary, I recommend that this paper be published but suggest that the authors reframe the narrative of the*

*paper around improving and characterizing the SIFT instrument, deemphasize the comparison with the PTR instrument, and bring a large portion of the impressive SI work into the main paper.*

Reviewer 2 also made the same point. In response, we moved the flow tube voltage, temperature, and carrier gas and sample gas flow optimization plots (now Fig. 2-5) to the main paper as those were the most thoroughly and systematically tested parameters. We also moved up the short discussion parts of the extended figure captions of the SI into the main paper, so that the optimization done by us is now the main focus of the manuscript. We moved the section about the SIFT-MS robustness from the comparison with the PTR-MS to its own section. We extended the section on the SIFT-MS humidity-dependence, also moving two graphs from the SI to the main paper as support (now Fig. 7 and 8). To reframe the narrative around the optimization of the SIFT-MS and minimize direct comparison with PTR-MS we changed text in the abstract, introduction, and conclusion and changed the title of the paper.

Examples:

P2 L5: "Here we present several improvements to a Voice 200 ultra SIFT-MS instrument to reduce background levels and enhance sensitivity. Increasing the sample gas flow to 125 sccm enables LODs at sub-ppb level, and the resulting humidity-dependence is overcome by calibrating for humidity as well. A comparison with a PTR-QMS 500 indicated that detection limits of the PTR-MS were an order of magnitude lower, whereas sensitivity was higher for SIFT-MS and its calibration was still more robust against humidity. Overall, SIFT-MS is a suitable, lower-cost and easy-to-use method for atmospheric trace gas measurements of more complex mixtures, even with isomers, at a varying humidity range."

P3 L22: "As mentioned above, the high limits of detection (LODs) for SIFT-MS can be an issue when measuring atmospheric trace gases, so we optimized the Voice 200 ultra SIFT-MS (Syft Technologies, New Zealand) to meet our requirements of LODs at sub-ppb level and systematically characterized the performance of the SIFT-MS under different humidity-conditions. Lastly, the instrument's performance was compared to the performance of a PTR-QMS 500 (Ionicon, Austria)."

P12 L18: "We successfully improved a purchased SIFT-MS to meet the requirements of sub-ppb atmospheric trace gas measurements. Hardware improvements like changing o-rings in the purchased instrument for materials with lower degassing, and exchanging the capillary in the inlet system with a VICI valve helped reduce the SIFT-MS background. Increasing the sample gas flow by a factor of 5 also improved sensitivity greatly, but made adjustments of the carrier gas flow, the flow tube voltage and temperature necessary. In total, we achieved a decrease of the SIFT-MS' LOD by a factor of 10."

*Reviewer 1: Specific comments: Section 3.1 seems to repeat a large amount of the material in Section 2.2. A description of the SIFT improvements should be in one or the other, but not both.*

We shortened Section 2.2 to make sure it contains only a description of what was done.

*Reviewer 1: P3 L7: ": : :usually a quadrupole MS". Do the authors have evidence of this? For SIFT this is true, but TOF has been the dominant technique for PTR for almost a decade now. This is not trivial, as the issues with how the authors characterize PTR are serious throughout the work.*

Changed to "...usually a quadrupole-MS for SIFT-MS and a time of flight-MS for PTR-MS,..." (P3 L6)

*Reviewer 1: P4 L16-18: The authors judge the success of their instrument modifications here by evaluating the background. Did they also evaluate response time, which is a critical parameter for ambient atmospheric measurements? PEEK may have a higher background, but its response time has been shown to be significantly better than stainless steel (coated and non). (Deming et al., 2018)*

See above, we usually do not observe lag times on the instruments after 30 s, and the silcosteel lines from the multi-port inlet to the flow tube are longer (approximately 15-20 cm) than the capillaries (approx.. 6-8 cm), so we do not think the capillary has such a big effect here. Plus, we shortened the silcosteel lines by using the needle valve instead of a capillary by approximately 10 cm.

*Reviewer 1: P10 L17: Mass-to-charge ratios should have a label as such everywhere in the paper. e.g. m/z 19 or m/Q 75*

We introduced this way of describing the ion in the methods section (P5 L16 "For the sake of simplicity, we will refer to the individual ions by m/z(reagent ion) / m/z(product ion) / analyte, e.g. 19 u / 33 u / methanol throughout the paper."), but the reviewer is right, we should at least have used the unit of the ions, so we added "u" to each m/z ratio to make clear it is an m/z value.

*Reviewer 1: Technical corrections: P4 L33: This is the methods section. Are the authors referring to a different methods section?*

Sorry, this was an error, we removed the "As described in the methods section" part – what we meant was described directly above.

*Reviewer 1: P5 L20: should be "it more strongly punishes a larger number of parameters"*

Corrected.

*Reviewer 1: P6 L25: "workdaily" I'm not familiar with this word.*

Corrected to "on each working day".

References: Breitenlechner, M., Fischer, L., Hainer, M., Heinritzi, M., Curtius, J., and Hansel, A.: PTR3: An Instrument for Studying the Lifecycle of Reactive Organic Carbon in the Atmosphere, Anal. Chem., 89, 5824-5831, 2017. Deming, B. L., Pagonis, D., Liu, X., Day, D. A., Talukdar, R., Krechmer, J. E., de Gouw, J. A., Jimenez, J. L., and Ziemann, P. J.: Measurements of delays of gas-phase compounds in a wide variety of tubing materials due to gas–wall interactions, Atmos. Meas. Tech., 12, 3453-3461, 2019. Krechmer, J., Lopez-Hilfiker, F., Koss, A., Hutterli, M., Stoermer, C., Deming, B., Kimmel, J., Warneke, C., Holzinger, R., Jayne, J., Worsnop, D., Fuhrer, K., Gonin, M., and de Gouw, J.: Evaluation of a New Reagent-Ion Source and Focusing Ion-Molecule Reactor for Use in Proton-Transfer-Reaction Mass Spectrometry, Anal. Chem., 90, 12011-12018, 2018. Lourenço, C., González-Méndez, R., Reich, F., Mason, N., and Turner, C.: A potential method for comparing instrumental analysis of volatile organic compounds using standards calibrated for the gas phase, Int. J. Mass spectrom., 419, 1-10, 2017. Prince, B. J., Milligan, D. B., and McEwan, M. J.: Application of selected ion flow tube mass spectrometry to real-time atmospheric monitoring, Rapid Commun.

Mass Spectrom., 24, 1763-1769, 2010. Sulzer, P., Hartungen, E., Hanel, G., Feil, S., Winkler, K., Mutschlechner, P., Haidacher, S., Schottkowsky, R., Gunsch, D., Seehauser, H., Striednig, M., Jürschik, S., Breiev, K., Lanza, M., Herbig, J., Märk, L., Märk, T. D., and Jordan, A.: A Proton Transfer Reaction-Quadrupole interface Time-Of-Flight Mass Spectrometer (PTR-QiTOF): High speed due to extreme sensitivity, Int. J. Mass spectrom., 368, 1-5, 2014. Yuan, B., Koss, A. R., Warneke, C., Coggon, M., Sekimoto, K., and de Gouw, J. A.: Proton-Transfer-Reaction Mass Spectrometry: Applications in Atmospheric Sciences, Chem. Rev., 117, 13187-13229, 2017.

---

## Author Comment (AC2) · 19 May 2020

*Reviewer 2: The paper presents results of extensive an thorough comparison of the current SIFT-MS instrument Voice 200 ultra with an older version of PTR-MS: QMS 500. Whilst the data are interesting, the results must not be interpreted as a test showing advantages and disadvantages of the current products on the market. The results are nevertheless based on honest and independent experimental work and should be published, as they do represent important contribution to validation of SIFT-MS in particular for VOC emissions from soils at different humidity. It is worthy of note that PTR-MS is used at 136 Td, some clear justification should be given for this*

[Figure]

*value. Perhaps a lower value would reduce fragmentation?*

Response: See P. 11 L 21: "The authors are aware that this increases fragmentation reactions, however, we found the settings to work well for humid samples: The formation of m/z = 37 u and water clusters of product ions is reduced substantially. Also, we reduced the risk of water and VOCs condensing in the inlet tubes by using the stated high inlet temperature and drift tube temperature. "

*Reviewer 2: The important realization is, that the background to a large degree originates from impurities in the instrument components. A very interesting finding relates to comparatively novel use of N2 carrier: "Nitrogen carrier gas lead to a higher sensitivity, but worse LODs and SNRs at 1 ppb and showed a higher humidity-sensitivity of the reagent ions, so we decided to use helium." Clearly more research needs to be done on the N2 carrier gas. "Final running conditions for the SIFT-MS were: 40 V, 140°C, 158 sccm Helium, and 100 sccm sample." This is a comparatively large proportion of sample, presumably following the work of Marvin Shaw, as indicated in section 2.2.*

Response: We were aware of Mavin Shaw's work, however, we mainly chose to test the increase in the sample gas flow, because we expected it to increase the amount of analyte in the flow tube and thus the number of product ions formed. But since we were inspired by his work, we put a statement in to acknowledge it:
P. 7 L. 1: "Several changes were applied to the SIFT-MS to improve its limit of detection, inspired by the optimizations done by Marvin Shaw (University of York, unpublished results), but considering different sample humidities:"

*Reviewer 2: Does the worse LOD for nitrogen mean a higher background? If so, is it*

*the purity of the nitrogen that is the problem?*

Response: Yes, it generally meant a higher background. However, since we already used nitrogen 6.0 with an additional scrubber, there is not much that one can improve here. Plus, the helium had the same quality and did not show so many impurities. The higher collisional cross section of nitrogen molecules might also just lead to an increased visibility of impurities, as it could increase ionization efficiency.
Included in P. 8 L. 16ff: "Further, humidity-sensitivity of the reagent ions was also higher with nitrogen carrier gas, as was instrument background. In both cases 6.0 quality gases were used and the nitrogen was even further purified with a filter, so that total amount of impurities should be similar for both gases. We thus attribute the higher background we observed with nitrogen to the higher collisional cross-section of nitrogen molecules compared to helium atoms, which might have caused a higher ionization efficiency of the impurities in the nitrogen and the instrument itself, basically increasing the visibility of the impurities by increasing the amount of ionized background analytes. To this, we also attribute the higher sensitivity we observed with nitrogen."

*Reviewer 2: Added P. 9 L 17 ff: "As in both cases 6.0 quality gases were used and the nitrogen was even further purified with a filter, the total amount of impurities should be similar and cannot explain the difference in instrument background. We thus attribute the higher background we observed with nitrogen to the higher collisional cross-section of nitrogen molecules compared to helium atoms, which might have caused a higher ionization efficiency of the impurities in the nitrogen and the instrument itself. To this, we also attribute the higher sensitivity we observed with nitrogen. The ion chemistry discussed in 3.2 would perhaps deserve better terminology and insight instead of "intensity loss" of m/z 33 and gain of 51 it would be better to mention, for example, occurence of three body association of protonated methanol with water. Was m/z 69*

*present at all for methanol at 140C?*

Response: On P. 9 L 20 ff: we exchanged the words "gain" and "loss" with "increase" and "decrease" and added a brief discussion what might happen to the ions in L 21 ff: "This could reflect either an increased association of water to protonated methanol in a three-body association involving a third collision partner M that takes up excess energy ($CH_3OH \cdot H^+ + H_2O + M \rightarrow CH_3OH \cdot H^+ \cdot H_2O + M^*$), or an increased ionization of methanol by $H_3O^+ \cdot H_2O$, where one water ligand is exchanged for methanol ($CH_3OH + H_3O^+ \cdot H_2O \rightarrow CH_3OH \cdot H_3O^+ + H_2O$) "
Discussion on whether m/z 69 for methanol is present, cf P. 9 L 24 ff: " $CH_3OH \cdot H^+ \cdot 2H_2O$ (m/z($H_3O^+$) = 69 u) could not be observed directly, as we used a mixed VOC standard and at this m/z, isoprene is also detected. A quick calculation of the isoprene signal we should see based on the isoprene signal we see at m/z($NO^+$) = 68 u showed us that most of the observed signal should be from isoprene and if at all only a minor amount of the methanol dihydrate ion should be present. For the exact calculation, please refer to the Supporting Information, S4.1."

*Reviewer 2: The discussion on acetaldehyde and proton affinity would also do with use of established terminology. The term "deionize" does not seem to be the best choice. What is likely to happen is a sequence of association reactions followed by final ligand switching to form $H_3O^+ \cdot (H_2O)_n$*
*$MH^+ \cdot H_2O_{(n)} + H_2O \rightarrow H_3O^+ \cdot H_2O_{(n)} + M$ So the charge moves back into the reagent ion system. See for example our early paper on formaldehyde Spanel et al. RCM 13, 1354 (1999) or on hydrocarbons, equation 7 in Spanel Smith IJMS 181 (1998) 1. "In accordance with Wilson et al. (2003), we conclude that a back reaction of the product ion with water might deionize the product ion to form a thermally colder reagent ion again and that this might correspond to the proton affinity of the compound" This is more to do with apparent $H_3O^+$ affinity than $H^+$ affinity.*

Response: This is the reaction we tried to describe by the word deionize, as essentially the analyte is not ionized anymore. We agree that it might not be the best wording and changed the paragraph to the following (P. 9 L 35):

"In accordance with Wilson et al. (2003), we conclude that a back-reaction of the product ion with water occurs by a ligand exchange of $H_3O^+ \cdot M$: The analyte M is exchanged by water again and thus not part of the ion anymore, leaving a thermally colder reagent ion behind: e.g. $CH_3CHO \cdot H_3O^+ + H_2O \rightarrow H_3O^+ \cdot H_2O + CH_3CHO$ (Spanel and Smith, 1998). This affinity to $H_3O^+$ should correspond to the proton affinity of the compound, as $H_3O^+$ is essentially a proton with one water ligand associated." (P. 9, L 30 ff). Also, "gain" and "loss" were exchanged for "decrease" and "increase" again.

*Reviewer 2: I agree with the other referee in that much of the data could be in the text and not in Suppl.*

Response: As also stated in our response to Reviewer 1, we moved the flow tube voltage, temperature, and carrier gas and sample gas flow optimization plots (now Fig. 2-5) to the main paper as those were the most thoroughly and systematically tested parameters. We also moved up the short discussion parts of the extended figure captions of the SI into the main paper, so that the optimization done by us is now the main part of the manuscript. We moved the section about the SIFT-MS robustness from the comparison with the PTR-MS to an own section and also extended the section on the SIFT-MS humidity-dependence a bit, also moving two graphs to the main paper from this part (now Fig. 7 and 8).

---

## Author Comment (AC3) · 29 May 2020

*Reviewer: This paper provides some interesting original research into the detection of adulteration of Olive Oils. It should be published but a significant revision is needed before this can be achieved as there are a number of issues that require clarification. [...]*

Reply: Thank you for your comments. However, we did not report research about the adulteration of Olive Oils in our paper, but demonstrate how to improve SIFT-MS for soil VOC emission measurements. So we think an error or a mix-up must have happened when uploading the review, and we are in contact with the editor to resolve the issue.